# ON TRANSPORTATION OF MINI-BATCHES:
# A HIERARCHICAL APPROACH

## ABSTRACT

Mini-batch optimal transport (m-OT) has been successfully used in practical applications that involve probability measures with a very high number of supports. The m-OT solves several smaller optimal transport problems and then returns the average of their costs and transportation plans. Despite its scalability advantage, the m-OT does not consider the relationship between mini-batches which leads to undesirable estimation. Moreover, the m-OT does not approximate a proper metric between probability measures since the identity property is not satisfied. To address these problems, we propose a novel mini-batching scheme for optimal transport, named *Batch of Mini-batches Optimal Transport* (BoMb-OT), that finds the optimal coupling between mini-batches and it can be seen as an approximation to a well-defined distance on the space of probability measures. Furthermore, we show that the m-OT is a limit of the entropic regularized version of the BoMb-OT when the regularized parameter goes to infinity. Finally, we present the new algorithms of the BoMb-OT in various applications, such as deep generative models and deep domain adaptation. From extensive experiments, we observe that the BoMb-OT achieves a favorable performance in deep learning models such as deep generative models and deep domain adaptation. In other applications such as approximate Bayesian computation, color transfer, and gradient flow, the BoMb-OT also yields either a lower quantitative result or a better qualitative result than the m-OT.

## 1 INTRODUCTION

Optimal transport (OT) (Villani, 2021; 2008; Peyré et al., 2019) has emerged as an efficient tool in dealing with problems involving probability measures. Under the name of Wasserstein distance, OT has been widely utilized to solve problems such as generative modeling (Arjovsky et al., 2017; Tolstikhin et al., 2018; Salimans et al., 2018; Genevay et al., 2018; Liutkus et al., 2019), barycenter problem (Ho et al., 2017; Li et al., 2020), and approximate Bayesian computation (Bernton et al., 2019a; Nadjahi et al., 2020). Furthermore, OT can also provide the most economical map of moving masses between probability measures, which is very useful in various tasks such as color transfer (Ferradans et al., 2014; Perrot et al., 2016), natural language procesing (Alvarez-Melis & Jaakkola, 2018), and domain adaptation (alignment) (Courty et al., 2016; Lee et al., 2019a; Xu et al., 2020), and graph processing (Titouan et al., 2019; Xu et al., 2019; Chen et al., 2020).

Although OT has attracted growing attention in recent years, a major barrier that prevents OT from being ubiquitous is its heavy computational cost. When the two probability measures are discrete with $n$ supports, solving the Wasserstein distance via the interior point methods has the complexity of $\mathcal{O}(n^3 \log n)$ (Pele & Werman, 2009), which is extremely expensive when $n$ is large. There are two main lines of works that focus on easing this computational burden. The first approach is to find a good enough approximation of the solution by adding an entropic regularized term on the objective function (Cuturi, 2013). Several works (Altschuler et al., 2017; Lin et al., 2019) show that the entropic approach can produce a $\varepsilon$-approximated solution at the same time reduces the computational complexity to $\mathcal{O}(n^2/\varepsilon^2)$. The second line of works named the "slicing" approach is based on the closed-form of Wasserstein distance in one-dimensional space, which has the computational complexity of order $\mathcal{O}(n \log n)$. There are various variants in this directions; i.e., (Bonneel et al., 2015; Deshpande et al., 2019; Kolouri et al., 2019; Nguyen et al., 2021a;b), these all belong to the family of sliced Wasserstein distances. Recently, some methods are proposed to combine the

dimensional reduction approach with the entropic solver of Wasserstein distance to inherit advantages from both directions (Paty & Cuturi, 2019; Muzellec & Cuturi, 2019; Lin et al., 2020a;b).

Although those works have reduced the computational cost of OT considerably, computing OT is nearly impossible in the big data settings where $n$ could be as large as few millions. In particular, solving OT requires to compute and store a $n \times n$ cost matrix that is impractical with current computational devices. Especially in deep learning applications, both supports of empirical measures and the cost matrix must be stored in a same device (e.g. a GPU) for automatic differentiation. This problem exists in all variants of OT including Wasserstein distance, entropic Wasserstein, and sliced Wasserstein distance. Therefore, it leads to the development of the mini-batches method for OT (Genevay et al., 2018; Sommerfeld et al., 2019), which we refer to as mini-batch OT loss. The main idea of the mini-batch method is to divide the original samples into multiple subsets (mini-batches), in the hope that each pair of subsets (mini-batches) could capture some structures of the two probability measures, meanwhile, the computing OT cost between two mini-batches is cheap due to a very small size of mini-batches. Then the overall loss is defined as the average of distances between pairs of mini-batches. This scheme was applied for many forms of Wasserstein distances (Deshpande et al., 2018; Bhushan Damodaran et al., 2018; Kolouri et al., 2018; Salimans et al., 2018), and was theoretically studied in the works of (Bellemare et al., 2017; Bernton et al., 2019b; Nadjahi et al., 2019). Recently, Fatras et al. (Fatras et al., 2020; 2021b) formulated this approach by giving a formal definition of the mini-batch OT loss, studying its asymptotic behavior, and investigating its gradient estimation properties. Despite being applied successfully, the current mini-batch OT loss does not consider the relationship between mini-batches and treats every pair of mini-batches the same. This causes undesirable effects in measuring the discrepancy between probability measures. First, the m-OT loss is shown to be an approximation of a discrepancy (the population m-OT) that does not preserve the metricity property, namely, this discrepancy is always positive even when two probability measures are identical. Second, it is also unclear whether this discrepancy achieves the minimum value when the two probability measures are the same. That naturally raises the question if we could propose a better mini-batching scheme to sort out these issues in order to improve the performance of the OT in practical applications.

**Contribution:** In this paper, we propose a novel mini-batching scheme for optimal transport, which is named as *Batch of Mini-batches Optimal Transport* (BoMb-OT). In particular, the BoMb-OT views every mini-batch as a point in the product space, then a set of mini-batches could be considered as an empirical measure. We now could employ the Kantorovich formulation between these two empirical measures in the product space as a discrepancy between two sets of mini-batches. In summary, our main contributions are two-fold:

1. First, the BoMb-OT could provide a more similar transportation plan to the original OT than the m-OT, which leads to a more meaningful discrepancy using mini-batches. In particular, we prove that the BoMb-OT approximates a well-defined metric on the space of probability measures, named population BoMb-OT. Furthermore, the entropic regularization version of population BoMb-OT could be employed as a generalized version of the population m-OT. Specifically, when the regularization parameter in the entropic population BoMb-OT goes to infinity, its value approaches the value of the population m-OT.

2. Second, we present the implementation strategy of the BoMb-OT and detailed algorithms in various applications in Appendix C. We then demonstrate the favorable performance of the BoMb-OT over the m-OT in two main applications that using optimal transport losses, namely, deep generative models and deep domain adaptation. Moreover, we also compare BoMb-OT to m-OT in other applications, such as sample matching, approximate Bayesian computation, color transfer, and gradient flow. In all applications, we also provide a careful investigation of the effects of two hyper-parameters of the mini-batching scheme, which are the number of mini-batches and the size of mini-batches, on the performance of the BoMb-OT and the m-OT.

**Organization:** The remainder of the paper is organized as follows. In Section 2, we provide backgrounds for optimal transport distances and the conventional mini-batching scheme (m-OT). In Section 3, we define the new mini-batching scheme for optimal transport distances, Batch of mini-batches Optimal Transport, and derive some of its theoretical properties. Section 4 benchmarks the proposed mini-batch scheme by extensive experiments on large-scale datasets, and followed by discussions in Section 5. Finally, proofs of key results and extra materials are in the supplementary.

**Notation:** For any probability measure $\mu$ on the Polish measurable space $(\mathcal{X}, \Sigma)$, we denote $\overset{\otimes m}{\mu}(m \geq 2)$ as the product measure on the product measurable space $(\mathcal{X}^m, \Sigma^m)$. For any $p \geq 1$, we define $\mathcal{P}_p(\mathbb{R}^N)$ as the set of Borel probability measures with finite $p$-th moment defined on a given metric space $(\mathbb{R}^N, \|.\|)$. To simplify the presentation, we abuse the notation by using both the notation $X^m$ for both the random vector $(x_1, \ldots, x_m) \in \mathcal{X}^m$ and the set $\{x_1, \ldots, x_m\}$, and we define by $P_{X^m} := \frac{1}{m}\sum_{i=1}^m \delta_{x_i}$ the empirical measure (the mini-batch measure) associated with $X^m$. For any set $X^n := \{x_1, \ldots, x_n\}$ and $m \geq 1$, we denote by $[X^n]^m$ the product set of $X^n$ taken $m$ times and $\binom{X^n}{m}$ the set of all $m$-element subsets of $X^n$.

## 2 BACKGROUND ON MINI-BATCH OPTIMAL TRANSPORT

In this section, we first review the definitions of the Wasserstein distance, the entropic Wasserstein, and the sliced Wasserstein. We then review the definition of the mini-batch optimal transport (m-OT).

### 2.1 WASSERSTEIN DISTANCE AND ITS VARIANTS

We first start with the definition of Wasserstein distance and its variants. Let $\mu$ and $\nu$ be two probability measures on $\mathcal{P}_p(\mathbb{R}^N)$. The Wasserstein $p$-distance between $\mu$ and $\nu$ is defined as follows:

$W_p(\mu, \nu) := \min_{\pi \in \Pi(\mu,\nu)} \left[ \mathbb{E}_{\pi(x,y)} \|x - y\|^p \right]^{\frac{1}{p}}$, where $\Pi(\mu, \nu) := \left\{ \pi : \int \pi dx = \nu, \int \pi dy = \mu \right\}$ is the set of transportation plans between $\mu$ and $\nu$.

The entropic regularized Wasserstein to approximate the OT solution (Altschuler et al., 2017; Lin et al., 2019) between $\mu$ and $\nu$ is defined as follows (Cuturi, 2013): $W_p^\tau(\mu, \nu) := \min_{\pi \in \Pi(\mu,\nu)} \left\{ \left[ \mathbb{E}_{\pi(x,y)} \|x - y\|^p \right]^{\frac{1}{p}} + \tau \mathrm{KL}(\pi | \mu \otimes \nu) \right\}$, where $\tau > 0$ is a chosen regularized parameter and KL denotes the Kullback-Leibler divergence.

Finally, the sliced Wasserstein (SW) (Bonnotte, 2013; Bonneel et al., 2015) is motivated by the closed-form of the Wasserstein distance in one-dimensional space. The formal definition of SW is: $SW_p(\mu, \nu) := \left[ \mathbb{E}_{\theta \sim \mathcal{U}(\mathbb{S}^{N-1})} W_p^p(\theta\sharp\mu, \theta\sharp\nu) \right]^{\frac{1}{p}}$, where $\mathcal{U}(\mathbb{S}^{N-1})$ denotes the uniform measure over the $(N-1)$-dimensional unit hypersphere and $\theta\sharp$ is the orthogonal projection operator on direction $\theta$.

### 2.2 MINI-BATCH OPTIMAL TRANSPORT

In this section, we first discuss the memory issue of large-scale optimal transport and challenges of dual solver. Then, we revisit the mini-batch OT loss that has been used in training deep generative models, domain adaptations, color transfer, and approximate Bayesian computation (Bhushan Damodaran et al., 2018; Genevay et al., 2018; Tolstikhin et al., 2018; Fatras et al., 2020; Bernton et al., 2019a). To ease the presentation, we are given $X^n := \{x_1, \ldots, x_n\}$, $Y^n := \{y_1, \ldots, y_n\}$ i.i.d. samples from $\mu$ and $\nu$ in turn. Let $\mu_n := \frac{1}{n}\sum_{i=1}^n \delta_{x_i}$ and $\nu_n := \frac{1}{n}\sum_{i=1}^n \delta_{y_i}$ be two corresponding empirical measures from the whole data set. Here, $n$ is usually large (e.g., millions) and each support in $X^n$, $Y^n$ can be a high dimensional data point (e.g. a high resolution image, video, etc).

**Memory issue of optimal transport:** Using an OT loss between $\mu_n$ and $\nu_n$ needs to compute and store a $n \times n$ cost matrix which has one trillion float entries (about 4 terabytes) when $n$ is about millions. Moreover, when dealing with deep neural networks, both support points and the cost matrix are required to be stored in the same memory (e.g., a GPU with 8 gigabytes memory) as a part of the computational graph for automatic differentiation. This issue applies to all variants of OT losses, such as Wasserstein distance, entropic Wasserstein, sliced Wasserstein distance. Therefore, it is *nearly impossible* to compute OT and its variants in large-scale applications.

**Challenges of stochastic dual solver**: Using stochastic optimization to solve the Kantorovich dual form is a possible approach to deal with large-scale OT, i.e. Wasserstein GAN (Arjovsky et al., 2017; Leygonie et al., 2019). However, the obtained distance has been shown to be very different from the original Wasserstein distance (Mallasto et al., 2019; Stanczuk et al., 2021). Using input convex neural networks is another choice to approximate the Brenier potential (Makkuva et al., 2020). Nevertheless, recent work (Korotin et al., 2021) has indicated that input convex neural networks are not sufficient (have limited power) in approximating the Brenier potential. Furthermore, both mentioned approaches are restricted in the choice of ground metric. In particular, $\mathcal{L}_1$ norm is used in

Wasserstein GAN to make the constraint of dual form into the Lipchitz constraint and $\mathcal{L}_2$ norm is for the existence of the Brenier potential.

**Mini-batch solution:** As a popular alternative approach for stable large-scale OT with flexible choices of the ground metric, the mini-batch optimal transport is proposed (Genevay et al., 2018; Sommerfeld et al., 2019) and has been widely used in various applications (Arjovsky et al., 2017; Deshpande et al., 2018; Sommerfeld et al., 2019; Bhushan Damodaran et al., 2018). In this approach, the original $n$ samples are divided into subsets (mini-batches) of size $m$, where $m$ is often the largest number that the computer can process, then an alternative solution of the original OT problem is formed by aggregating these smaller OT solutions from mini-batches.

We now state an adapted definition of mini-batch OT (m-OT) scheme that was formulated in (Fatras et al., 2020), including its two key parts: its transportation cost and its transportation plan.

**Definition 1** (Empirical m-OT). *For $p \geq 1$ and integers $m \geq 1$, and $k \geq 1$, let $d : \mathcal{P}_p(\mathcal{X}) \times \mathcal{P}_p(\mathcal{X}) \to [0, \infty)$ be a function, i.e., $\{W_p, W_p^\tau, SW_p\}$. Then, the mini-batch OT (m-OT) loss and the transportation plan, a $n \times n$ matrix, between $\mu_n$ and $\nu_n$ are defined as follows:*

$$\widehat{U}_d^{k,m}(\mu_n, \nu_n) := \frac{1}{k^2} \sum_{i=1}^k \sum_{j=1}^k d(P_{X_i^m}, P_{Y_j^m}); \quad \widehat{\pi}_k^m(\mu_n, \nu_n) := \frac{1}{k^2} \sum_{i=1}^k \sum_{j=1}^k \pi_{X_i^m, Y_j^m}, \quad (1)$$

*where $X_i^m$ is sampled i.i.d from $\binom{X^n}{m}$, $Y_i^m$ is sampled i.i.d from $\binom{Y^n}{m}$, and the transport plan $\pi_{X_i^m, Y_j^m}$, where its entries equal zero except those indexed of samples $X_i^m \times Y_j^m$, is the transportation plan when $d(P_{X_i^m}, P_{Y_j^m})$ is an optimal transport metric.*

The above definition was generalized to the two original measures as follows in (Fatras et al., 2020):

**Definition 2** (Population m-OT). *Assume that $\mu$ and $\nu$ are two probability measures on $\mathcal{P}_p(\mathcal{X})$ for given positive integers $p \geq 1$, $m \geq 1$, and $d : \mathcal{P}_p(\mathcal{X}) \times \mathcal{P}_p(\mathcal{X}) \to [0, \infty)$ be a given function. Then, the population mini-batch OT (m-OT) discrepancy between $\mu$ and $\nu$ is defined as follows:*

$$U_d^m(\mu, \nu) := \mathbb{E}_{(X^m, Y^m) \sim \otimes^m \mu \otimes \otimes^m \nu} d(P_{X^m}, P_{Y^m}). \quad (2)$$

**Issues of the m-OT:** From Definition 1, the m-OT treats every pair of mini-batches the same by taking the average of the m-OT loss between any pair of them for both transportation loss and transportation plan. This treatment has some issues. First, a mini-batch is a sparse representation of the true distribution and two sparse representations of the same distribution could be very different from each other. Hence, a mini-batch from $X^m$ would prefer to match to certain mini-batches of $Y^m$, rather than treating every mini-batch of $Y^m$ equally. For example, each mini-batch has one datum, then each term in the population m-OT now is the ground metric of the OT cost, the population m-OT degenerates to $\mathbb{E}[d(X^m, Y^m)]$ for independent $X^m$ and $Y^m$. This treatment also leads to an uninformative transportation plan shown in Figure 15, which is followed by a less meaningful transportation cost. Second, although it has been proved that the population m-OT is symmetric and positive (Fatras et al., 2020), for the same reason it does not vanish when two measures are identical.

## 3 BATCH OF MINI-BATCHES OPTIMAL TRANSPORT

To address the issues of m-OT, in this section, we propose a novel mini-batch scheme, named *batch of mini-batches OT (BoMb-OT)*. We first demonstrate the improvement of the BoMb-OT to the m-OT and discuss the practical usage of the BoMb-OT. Then, we prove that the BoMb-OT loss is an approximation of a well-defined metric in the space of Borel probability measures. Finally, we discuss the entropic version of the population BoMb-OT (population eBoMb-OT) admits the population m-OT as a special case where the entropic regularization goes to infinity in Appendix B. To simplify the presentation, we assume throughout this section that $\mu$ and $\nu$ are continuous probability measures in $\mathcal{P}_p(\mathcal{X})$ for some given $p \geq 1$ and $\mathcal{X} \subset \mathbb{R}^N$ where $N \geq 1$. Furthermore, $d : \mathcal{P}_p(\mathcal{X}) \times \mathcal{P}_p(\mathcal{X}) \to [0, \infty)$ is a given divergence between probability measures in $\mathcal{P}_p(\mathcal{X})$.

### 3.1 DEFINITION OF BOMB-OT AND ITS PROPERTIES

Intuitively, a mini-batch scheme in OT can be decomposed into two steps: (1) Solving local OT between every pair of mini-batches from samples of two original measures, (2) Combining local OT

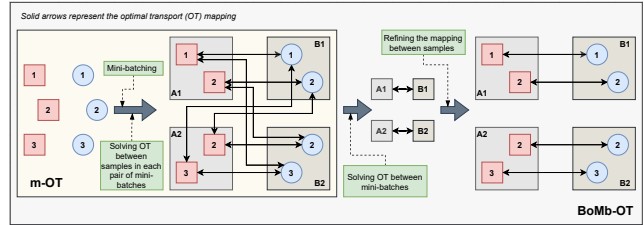

Figure 1: Visualization of the m-OT and the BoMb-OT in providing a mapping between samples.

solutions into a global solution. As discussed above, the main problem with the m-OT lies in the second step, all pairs of mini-batches play the same role in the total transportation loss, meanwhile, a mini-batch from $X^n$ would prefer to match some certain mini-batches from $Y^n$. To address the limitation, we enhance the second step by considering the optimal coupling between mini-batches. By doing so, we could define a new mini-batch scheme that is able to construct a good global mapping between samples that also leads to a meaningful objective loss.

Let $X_1^m, X_2^m, \ldots, X_k^m$ be mini-batches that are sampled with replacement from $[X^n]^m$ and $Y_1^m, Y_2^m, \ldots, Y_k^m$ be mini-batches that are sampled with replacement from $[Y^n]^m$. The batch of mini-batches optimal transport is defined as follows:

**Definition 3** (Batch of mini-batches). *Assume that $p \geq 1$, $m \geq 1$, $k \geq 1$ are positive integers and let $d : \mathcal{P}_p(\mathcal{X}) \times \mathcal{P}_p(\mathcal{X}) \to [0, \infty)$ be a given function (e.g., $\{W_p, W_p^\tau, SW_p\}$). The BoMb-OT loss and the BoMb-OT transportation plan between probability measures $\mu_n$ and $\nu_n$ are given by:*

$$\widehat{\mathcal{D}}_d^{k,m}(\mu_n, \nu_n) := \min_{\gamma \in \Pi(\overset{\otimes m}{\mu_k}, \overset{\otimes m}{\nu_k})} \sum_{i=1}^k \sum_{j=1}^k \gamma_{ij} d(P_{X_i^m}, P_{Y_j^m}); \quad \widehat{\pi}_k^m(\mu_n, \nu_n) := \sum_{i=1}^k \sum_{j=1}^k \bar{\gamma}_{ij} \pi_{X_i^m, Y_j^m},$$

*where $\overset{\otimes m}{\mu_k} := \frac{1}{k} \sum_{i=1}^k \delta_{X_i^m}$ and $\overset{\otimes m}{\nu_k} := \frac{1}{k} \sum_{j=1}^k \delta_{Y_j^m}$ are two empirical measures defined on the product space via mini-batches (measures over mini-batches), $\bar{\gamma}$ is a $k \times k$ optimal transport plan between $\overset{\otimes m}{\mu_k}$ and $\overset{\otimes m}{\nu_k}$, and $\pi_{X_i^m, Y_j^m}$ is defined as in Definition 1.*

Compared to the m-OT, the BoMb-OT considers an additional OT between two measures on mini-batches for combining mini-batch losses. The improvement of the BoMb-OT over the m-OT is illustrated in Figure 1 in which we assume that we can only solve $2 \times 2$ OT problems and we need to deal with the OT problems with two empirical measures of 3 supports. The optimal transportation plan is to map $i$-th square to $i$-th circle. Assume that there are four mini-batches: A1 and A2 from $\mathcal{X}$ and B1 and B2 from $\mathcal{Y}$. The weight for the m-OT loss between A1 and B2 equals $\frac{1}{4}$, meanwhile, the map from A1 to B2 is unuseful. In this toy example, A1 should be mapped to B1 and A2 is for B2. It is also the solution that the BoMb-OT tries to obtain by re-weighting pairs of mini-batches through a transportation plan between two measures over mini-batches.

**Practical usage of the BoMb-OT:** There are three types of applications that the BoMb-OT can be utilized. The first one is *gradient-based* applications (deep learning) such as deep generative models, deep domain adaption, and gradient flow. In these applications, the goal is to estimate the gradient of a parameter of interest (a neural network) with an OT objective function. The second one is mapping-based applications, namely, we aim to obtain a transportation map between samples. Examples of mapping-based applications include color transfer, domain adaptation, and sample matching. Finally, the third type of applications is *discrepancy-based* applications where we need to know the discrepancy between two measures, e.g., approximate Bayesian computation, and searching problems. We now discuss the implementation of BoMb-OT in each scenario.

*Gradient-based (deep learning) applications*: For a better presentation, we assume that a system with a GPU (graphics processing unit) is used in the training process. We further assume that the GPU has enough space to store the neural net, the $2 \times m$ supports of two mini-batch measures, and the $m \times m$ cost matrix. The algorithm of the BoMb-OT consists of three steps: *Forwarding*, *Solving $k \times k$ OT*, and *Re-forwarding and Backwarding*. In the first step, $k^2$ OT problems of size $m \times m$ are computed in turn on GPU to obtain the $k \times k$ cost matrix which indicates the transportation cost between each pair of mini-batch measures. We recall that it is only enough memory for computing $m \times m$ OT on GPU at a time. After that, an OT problem with the found $k \times k$ cost matrix is solved

to find the coupling between mini-batches. Here, $k$ can be much larger than $m$ since the $k \times k$ OT can be solved on computer memory (RAM - Random-access memory), which is bigger than GPU's memory. We would like to emphasize that in the first two steps, automatic differentiation is not required. In the last step, we do re-forwarding and do backwarding (back-propagation) to obtain the gradients of each pair of mini-batches, then using the coupling in the second step to aggregate them into the final gradient signal of the parameter of interest. The detailed algorithms of the BoMb-OT for deep generative model and deep domain adaption are respectively described in Algorithm 1 in Appendix C.1 and Algorithm 2 in Appendix C.2. Compared to the m-OT, the BoMb-OT only costs an additional forwarding and an additional $k \times k$ OT problem that is not expensive since they do not require to create and store any computational graphs.

*Mapping-based applications*: In this type of application, the BoMb-OT also consists of three steps like in gradient-based applications. However, in the last step, the transportation plan (the mapping) is aggregated instead of the gradient. As an example, we present the BoMb-OT algorithm for color transfer in Algorithm 3 in Appendix C.3. We would like to emphasize that the last step (re-forwarding) can be removed by storing all $k^2$ mini-batch transportation plans of size $m \times m$ which can be represented by a sparse matrix of size $n \times n$ for saving memory.

*Discrepancy-based applications*: The BoMb-OT needs only two steps in this scenario, namely, forwarding and solving $k \times k$ OT. Since we do not need to re-estimate any statistics from mini-batches, the final value of the BoMb-OT can be evaluated at the end of the second step without the re-forwarding step. Similar to the previous two types of applications, we present the BoMb-OT algorithm for approximate Bayesian computation in Algorithm 5 in Appendix C.4. Compared to the m-OT, the BoMb-OT needs only an additional $k \times k$ OT here.

**Choosing $k$ and $m$**: In practice, we want to have as large as possible values of $k$ and $m$. The mini-batch size $m$ is often chosen to be the largest value of the computational memory that can be stored. For choosing $k$, there are two practical cases. *Two (multiples) computational memories -* this is the case of modern computational systems that have at least a CPU (central processing unit) with RAM and a GPU with its corresponding memory. As discussed in the practical usage of the BoMb-OT, OT problems between mini-batch measures of size $m \times m$ are solved on the GPU for high computational speed in estimating the parameter of interest or other statistics. On the other hand, the $k \times k$ OT problems between measures over mini-batches can be computed on the CPU's memory. So, $k$ can be chosen larger than $m$. *One (shared) computational memory -* in this case, the largest value of $k$ equals to $m$. Note that, smaller values of $k$ and $m$ can still be used for faster computation. We would like to recall that $k$ is usually set to 1 in almost all recent applications of OT (Genevay et al., 2018; Bhushan Damodaran et al., 2018; Tolstikhin et al., 2018; Bernton et al., 2019a). In our experiments, we will demonstrate that increasing $k$ could improve the performance.

**Computational complexity of the BoMb-OT:** The computational complexity of BoMb-OT depends on a particular choice of divergence $d$. In the experiments, we specifically consider three choices of $d \in \{W_p, W_p^\tau, SW_p\}$ where $\tau > 0$ is a chosen regularized parameter. Here, we only discuss the case when $d = W_p$, the discussion of other choices of $d$ is in Appendix B. When $d$ is the entropic Wasserstein distance, for each pair $X_i^m$ and $Y_j^m$, the computational complexity of approximating $d(P_{X_i^m}, P_{Y_j^m})$ via the Sinkhorn algorithm is of order $\mathcal{O}(m^2/\varepsilon^2)$ where $\varepsilon > 0$ is some desired accuracy (Lin et al., 2019). Hence, the computation cost of $k^2$ OT transport plans from $k^2$ pairs of mini-batches is of order $\mathcal{O}(k^2 m^2/\varepsilon^2)$ when using entropic regularization (see Appendix B for the definition). With another OT cost within $k^2$ pairs, the total computational complexity of computing the BoMb-OT is $\mathcal{O}(k^2(m^2+1)/\varepsilon^2)$. It means that the computational complexity of computing the BoMb-OT is comparable to the m-OT, which is $\mathcal{O}(k^2 m^2/\varepsilon^2)$.

**The BoMb-OT's transportation plan:** We discuss in detail the sparsity of BoMb-OT's transportation plan in Appendix B. To visualize the BoMb-OT's plan and compare it with the m-OT's plan, we carry out a simple experiment on two measures of 10 supports in Figure 15. The details of the experiment and discussion are given in Appendix D.6. From this example, the BoMb-OT provides a more accurate transportation plan than the m-OT with various choices of $m$ and $k$.

## 3.2 METRIC APPROXIMATION OF THE BOMB-OT LOSS

In this section, we show that the BoMb-OT is an approximation of a well-defined metric, named *population BoMb-OT*, in the space of probability measures. To ease the ensuing discussion, we first define that metric as follows:

**Definition 4** (Population BoMb-OT). *Let $\mu$ and $\nu$ be two probability measures on $\mathcal{P}_p(\mathcal{X})$, for $p \geq 1$ is a positive number. The* population BoMb-OT *between probability measures $\mu$ and $\nu$ is defined as:*

$$\mathcal{D}_d^m(\mu, \nu) := \inf_{\gamma \in \Pi(\overset{\otimes m}{\mu}, \overset{\otimes m}{\nu})} \mathbb{E}_{(X^m, Y^m) \sim \gamma}[d(P_{X^m}, P_{Y^m})].$$

Different from the m-OT in Definition 2, the optimal plan between two distributions on product spaces is crucial to guarantee that the population BoMb-OT is a well-defined metric in the space of probability measures.

**Theorem 1** (Metric property of population BoMb-OT). *Assume that $d$ is an invariant metric under permutation on $\mathcal{X}^m$ and the function $d(P_{X^m}, P_{Y^m})$ is continuous in terms of $X^m, Y^m \in \mathcal{X}^m$. Then, the population BoMb-OT is a well-defined metric in the space of probability measures.*

The proof of Theorem 1 is in Appendix A. The assumption of Theorem 1 is mild and satisfied when $d$ is (sliced) Wasserstein metrics of order $p$, i.e., $d \in \{W_p, SW_p\}$.

Our next result shows the approximation error between the BoMb-OT loss and the population BoMb-OT distance. We provide the following result with the approximation error when $d \in \{W_p, SW_p\}$.

**Theorem 2** (Population BoMb-OT). *Assume that $p \geq 1$, $\mathcal{X}$ is a compact subset of $\mathbb{R}^N$, and all the possible mini-batches are considered. Then, we have (i) $\left|\widehat{\mathcal{D}}_d^{k,m}(\mu_n, \nu_n) - \mathcal{D}_d^m(\mu, \nu)\right| = \mathcal{O}_P\left(m^{1-\frac{1}{p}}/n^{\frac{1}{N}}\right)$ when $d \equiv W_p$; (ii) $\left|\widehat{\mathcal{D}}_d^{k,m}(\mu_n, \nu_n) - \mathcal{D}_d^m(\mu, \nu)\right| = \mathcal{O}_P\left(m^{1-\frac{1}{p}}/n^{\frac{1}{2}}\right)$ when $d \equiv SW_p$ and $k$ is the number of mini-batches.*

The proof of Theorem 2 is in Appendix A. We would like to remark that the dependency of the sample complexity of the BoMb-OT on $N$ is necessary when $d \equiv W_p$. It is due to the inclusion of the additional optimal transport in the BoMb-OT. On the other hand, the curse of dimensionality of the BoMb-OT loss does not happen when $d \equiv SW_p$. Furthermore, the choice $d \equiv SW_p$ improves not only the sample complexity but also the computational complexity of the BoMb-OT. In particular, in Appendix B, we demonstrate that the computational complexity of the BoMb-OT is $\mathcal{O}(k^2 m \log m)$ when $d \equiv SW_p$. Another potential scenario that the BoMb-OT does not have the curse of dimensionality is when we consider its entropic regularized version in equation 4. The entropic optimal transport had been shown to have sample complexity at the order $n^{-\frac{1}{2}}$ (Mena & Weed, 2019). In the case of the entropic regularized BoMb-OT (eBoMb-OT) (see equation (4) in Appendix B), when the regularized parameter $\lambda$ is infinity, namely, the m-OT, the result of (Fatras et al., 2020; 2021b) already established the sample complexity $n^{-\frac{1}{2}}$. However, for a general value of the regularized parameter, it is unclear whether we still have this sample complexity $n^{-\frac{1}{2}}$ of the eBoMb-OT. We leave this question for the future work.

## 4 EXPERIMENTS

In this section, we demonstrate the favorable performance of BoMb-OT compared to m-OT in three discussed types of applications, namely, *gradient-based, mapping-based, and value-based* applications. For gradient-based applications, we run experiments on deep generative model (Genevay et al., 2018; Deshpande et al., 2018), deep domain adaptation (Bhushan Damodaran et al., 2018). Experiments on color transfer (Rabin et al., 2014; Ferradans et al., 2014) are conducted as the example for mapping-based applications. Lastly, we present results on approximate Bayesian computation (Bernton et al., 2019a) which is a value-based application. In the main text of the paper, we only report and discuss the obtained experimental results, the details of applications and their algorithms are given in Appendix C. In Appendix D, we provide detailed experimental results of discussed applications including visualization and computational time. Moreover, we also carry out experiments on gradient flow (Santambrogio, 2017) and visualize mini-batch transportation matrices of the m-OT and the BoMb-OT. From all experiments, we observe that the BoMb-OT performs better than m-OT consistently. The detailed settings of our experiments including neural network architecture, hyper-parameters, and evaluation metrics are in Appendix E. We would like to recall that $k$ is the number of

Table 1: Comparison between the BoMb-OT and the m-OT on deep generative models. On the MNIST dataset, we evaluate the performances of generators by computing approximated Wasserstein-2 while we use FID score (Heusel et al., 2017) on CIFAR10 and CelebA.

| Dataset | $k$ | m-OT($W_2^\tau$) | BoMb-OT($W_2^\tau$) | eBoMb-OT ($W_2^\tau$) | m-OT($SW_2$) | BoMb-OT($SW_2$) | eBoMb-OT ($SW_2$) |
|---|---|---|---|---|---|---|---|
| MNIST | 1 | 28.12 | 28.12 | 28.12 | 37.57 | 37.57 | 37.57 |
| | 2 | 27.88 | **27.53** | 27.56 | 36.01 | **35.27** | 35.88 |
| | 4 | 27.6 | 27.42 | **27.41** | 35.18 | **34.19** | 34.85 |
| | 8 | 27.36 | **27.1** | 27.25 | 34.33 | **33.17** | 34.00 |
| CIFAR10 | 1 | 78.34 | 78.34 | 78.34 | 80.51 | 80.51 | 80.51 |
| | 2 | 76.20 | 75.59 | **74.25** | 67.86 | 65.22 | **62.80** |
| | 4 | 76.01 | 74.60 | **74.12** | 62.30 | 62.11 | **58.78** |
| | 8 | 75.22 | 74 | **73.33** | 59.68 | 58.94 | **53.44** |
| CelebA | 1 | 54.16 | 54.16 | 54.16 | 90.33 | 90.33 | 90.33 |
| | 2 | 52.85 | 52.49 | **51.53** | 82.45 | 78.66 | **74.48** |
| | 4 | 52.56 | 51.71 | **50.55** | 73.06 | 72.37 | **72.19** |
| | 8 | 51.92 | 51.18 | **49.63** | 71.95 | 69.3 | **68.52** |

Table 2: Comparison between two mini-batch schemes on deep domain adaptation on digits datasets. We vary the number of mini-batches $k$ and report the classification accuracy on the target domain.

| Scenario | $k$ | $m$ | Number of epochs | m-OT | BoMb-OT | Improvement | m-UOT | eBoMb-UOT | Improvement |
|---|---|---|---|---|---|---|---|---|---|
| SVHN to MNIST | 8 | 50 | 80 | 92 | 93.74 | +1.74 | 98.76 | 98.81 | +0.05 |
| | 16 | 50 | 160 | 93.06 | 94.32 | +1.26 | 98.83 | 98.85 | +0.02 |
| | 32 | 50 | 320 | 93.09 | **95.59** | +2.40 | 98.83 | 98.90 | +0.07 |
| USPS to MNIST | 8 | 25 | 80 | 95.86 | 96.16 | +0.30 | 98.46 | 98.59 | +0.13 |
| | 16 | 25 | 160 | 96.04 | 96.48 | +0.44 | 98.36 | 98.7 | +0.34 |
| | 32 | 25 | 320 | 96.22 | 96.71 | +0.49 | 98.43 | 98.75 | +0.32 |

mini-batches and $m$ is the mini-batch size. In our experiments, we create mini-batches by sampling without replacement from supports of the original empirical measures.

### 4.1 DEEP GENERATIVE MODEL

We now show the deep generative model result on MNIST (LeCun et al., 1998), CIFAR10 (32x32) (Krizhevsky, 2009), and CelebA (64x64) (Liu et al., 2015) using different mini-batch schemes with two OT losses for $d$: $SW_2$ (Deshpande et al., 2018) and $W_2^\tau$ (Genevay et al., 2018). According to Table 1, the BoMb-OT always gives lower quantitative metrics than the m-OT on all datasets with all choices of the number of mini-batches $k$ and mini-batch ground metric $d$. Interestingly, the eBoMb-OT which is the entropic regularization version of BoMb-OT (defined in Appendix B) is better than the BoMb-OT on CIFAR10, CelebA, and one setting on MNIST. The explanation for this phenomenon is that the eBoMb-OT loss is smoother than the BoMb-OT due to the entropic regularization (Genevay et al., 2018). From the table, we also observe that increasing the number of mini-batches $k$ improves the result of generators that are trained with the same number of stochastic gradient updates. It suggests that the gradient of an OT loss should not be estimated by only 1 pair of mini-batches like the way it has been implementing in practice. Also, based on the experimental results, we recommend practitioners to consider using an additional transportation problem between measures over mini-batches when the number of mini-batches $k > 1$.

### 4.2 DEEP DOMAIN ADAPTATION

In this section, we adapt two digits datasets SVHN (Netzer et al., 2011) and USPS (Hull, 1994) to MNIST (LeCun et al., 1998). Details about architectures of neural networks and hyper-parameters settings are given in Appendix E.2. The classification results are illustrated in Table 2. The BoMb-OT and eBoMb-UOT always achieve better results than their counterparts for all choices of $k$. As $k$ increases, the gap between two methods tends to become larger in both datasets. This demonstrates the effectiveness of our scheme, especially when dealing with a large number of mini-batches. We also observe that the classification accuracy on the target domain also increases as $k$ changes from 8 to 32 and keeping the same number of stochastic gradient updates for deep neural networks. Thus, we suggest that DeepJDOT (Bhushan Damodaran et al., 2018) and its variants should be implemented with the BoMb-OT's strategy which is presented in Algorithm 2 in Appendix C.2.

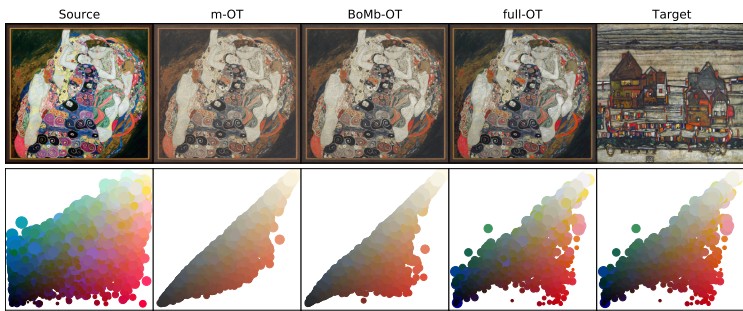

Figure 2: The transferred images of the m-OT and the BoMb-OT with k=10, m=10, and the full-OT from the most left source image to the most right target image.

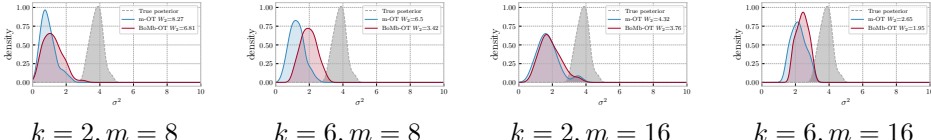

| $k = 2, m = 8$ | $k = 6, m = 8$ | $k = 2, m = 16$ | $k = 6, m = 16$ |

Figure 3: Illustration of approximated posteriors from ABC with the m-OT, the BoMb-OT, and the true posterior via kernel density estimation. The Wasserstein-2 distances from the approximated posteriors to the true posterior are given next to the label of corresponding mini-batch methods.

## 4.3 COLOR TRANSFER

Mini-batch OT has made color transfer to be able to transform color between two images that contain millions of pixels (Fatras et al., 2021b). In this section, we show that our new mini-batch strategy can improve further the quality of the color transfer. The details of the application and algorithm are given in Appendix C.3. The full experimental results including qualitative images and their color palettes with different choices of $k$ and $m$ are presented in Appendix D.3. Based on the experiment, we observe that the BoMb-OT provides a better barycentric mapping for color transfer than m-OT with every setting of $k$ and $m$. In this main paper, we show a color transfer result in Figure 2. We can observe that the color of the transferred image of the BoMb-OT is more similar to the target image than one of the m-OT. The color palettes in Appendix C.3 also reinforce this claim.

## 4.4 APPROXIMATE BAYESIAN COMPUTATION (ABC)

As choices of sample acceptance rejection criteria, we compare the m-OT and the BoMb-OT in ABC. We present the detail of the application and the algorithm in Appendix C.4. The setting of the model and the full results are given in Appendix D.4. We compare the m-OT and the BoMb-OT with various choices of $k$ and $m$. We compare them by plotting their approximated posteriors and computing the Wasserstein distance between approximated posteriors to the true posterior. According to the experiments, the BoMb-OT always yield lower Wasserstein distances than the m-OT. Here, we show some results in Figure 3. From the figure, we can see that approximated posteriors from the BoMb-OT are closer to the true posterior than the m-OT. Moreover, we observe that increasing the value of $k$ and $m$ improves the quality of the approximated posteriors from both mini-batch schemes considerably. Based on the fact that Wasserstein ABC is implemented with $k = 1$ (Bernton et al., 2019a), we suggest that the BoMb-OT should be used with $k$ is greater than 1 in ABC.

## 5 CONCLUSION

In the paper, we have presented a novel mini-batching scheme for optimal transport, named Batch of Mini-batches Optimal Transport (BoMb-OT). The idea of the BoMb-OT is to consider the optimal transport problem on the space of mini-batches with a Wasserstein-types ground metric. We prove that the BoMb-OT is an approximation of a valid distance between probability measures and its entropic regularized version, eBoMb-OT, is the generalization of the conventional mini-batch optimal transport. More importantly, we have shown that the BoMb-OT and the eBoMb-OT can be implemented efficiently and they have more favorable performance than the m-OT in various applications of optimal transport. For future work, we could consider another extension of the BoMb-OT by changing the local OT to the Gromov-Wasseretein (GW) and unbalanced OT (UOT).

**Reproducibility Statement:** All datasets that we used in the paper are published and they are easy to find on the Internet. Source codes and instruction for our experiments are provided in the supplementary of the paper. The details of experimental settings, computational infrastructure, and other used public libraries are given in Appendix E.

**Ethics Statement:** The paper investigates a fundamental practical problem of optimal transport in machine learning and deep learning. Hence, we do not foresee any ethical issues of the paper that can be discussed.

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

# Supplement to "On Transportation of Mini-batches: A Hierarchical Approach"

In this supplementary material, we collect several proofs and remaining materials that were deferred from the main paper. In Appendix A, we provide the proofs of the main results in the paper. We present additional materials including discussions about computation complexity of the BoMb-OT, the connection of the BoMb-OT to hierarchical optimal transport, entropic regularization of BoMb-OT, sparsity of the BoMb-OT, and extension with non-OT mini-batch metrics in Appendix B. Furthermore, we provide a description of applications and the BoMb-OT algorithms in those applications in Appendix C. Additional results of presented experiments in the main text in Appendix D, and their corresponding settings in Appendix E.

## A   PROOFS

In this appendix, we give detailed proofs of theorems that are stated in the main paper.

### A.1   PROOF OF THEOREM 1

We first prove that for any probability measures $\mu$ and $\nu \in \mathcal{P}_p(\Theta)$, there exists an optimal transportation plan $\gamma^*$ such that

$$\mathcal{D}_d^m(\mu, \nu) = \mathbb{E}_{(X^m, Y^m) \sim \gamma^*}[d(P_{X^m}, P_{Y^m})]. \tag{3}$$

From the definition of the BoMb-OT, there is a sequence of transportation plans $\gamma_n \in \Pi(\overset{\otimes m}{\mu}, \overset{\otimes m}{\nu})$ such that

$$\mathbb{E}_{(X^m, Y^m) \sim \gamma_n}[d(P_{X^m}, P_{Y^m})] \to \mathcal{D}_d(\mu, \nu)$$

as $n \to \infty$. Since $\Pi(\overset{\otimes m}{\mu}, \overset{\otimes m}{\nu})$ is compact in the weak* topology (Villani, 2008), $\gamma_n$ weakly converges to some $\gamma^* \in \Pi(\overset{\otimes m}{\mu}, \overset{\otimes m}{\nu})$. Since $d(P_{x^m}, P_{y^m})$ is continuous in terms of $x^m, y^m \in \mathcal{X}^m$, an application of Portmanteau theorem leads to

$$\lim_{n \to \infty} \mathbb{E}_{(X^m, Y^m) \sim \gamma_n}[d(P_{X^m}, P_{Y^m})] \geq \mathbb{E}_{(X^m, Y^m) \sim \gamma^*}[d(P_{X^m}, P_{Y^m})].$$

Putting the above results together, we obtain

$$\mathcal{D}_d^m(\mu, \nu) = \mathbb{E}_{(X^m, Y^m) \sim \gamma^*}[d(P_{X^m}, P_{Y^m})].$$

Therefore, there exists an optimal transportation plan $\gamma^*$ such that equation 3 holds.

We now proceed to prove that the BoMb-OT is a well-defined metric in the space of Borel probability measures. First, we demonstrate that $\mathcal{D}_d(\mu, \nu) = 0$ if and only if $\mu = \nu$. In fact, from the definition of $\mathcal{D}_d(.,.)$, if we have two probability measures $\mu$ and $\nu$ such that $\mathcal{D}(\mu, \nu) = 0$, we find that

$$\inf_{\gamma \in \Pi(\overset{\otimes m}{\mu}, \overset{\otimes m}{\nu})} \mathbb{E}_{(X^m, Y^m) \sim \gamma}[d(P_{X^m}, P_{Y^m})] = 0.$$

Since $d$ is a metric, it implies that there exists transportation plan $\gamma^* \in \Pi(\overset{\otimes m}{\mu}, \overset{\otimes m}{\nu})$ such that $P_{X^m} = P_{Y^m}$ $\gamma^*$-almost everywhere. It demonstrates that for each $x$, there exists a permutation $\sigma_x$ of $\{1, \ldots, m\}$ such that $x_i = y_{\sigma_x(i)}$ for all $i \in [m]$. Now, for any test function $f : \mathcal{X} \to \mathbb{R}$, we have

$$\int_{\mathcal{X}^m} \prod_{i=1}^m f(x_i) d\overset{\otimes m}{\mu}(x) = \int_{\mathcal{X}^m \times \mathcal{X}^m} \prod_{i=1}^m f(x_i) d\gamma^*(x, y) = \int_{\mathcal{X}^m \times \mathcal{X}^m} \prod_{i=1}^m f(y_{\sigma_x(i)}) d\gamma^*(x, y)$$

$$= \int_{\mathcal{X}^m \times \mathcal{X}^m} \prod_{i=1}^m f(y_i) d\gamma^*(x, y)$$

$$= \int_{\mathcal{X}^m} \prod_{i=1}^m f(y_i) d\overset{\otimes m}{\nu}(y).$$

Since $\int_{\mathcal{X}^m} \prod_{i=1}^m f(x_i) d\overset{\otimes m}{\mu}(x) = \left(\int_{\mathcal{X}} f(x_1) d\mu(x_1)\right)^m$ and $\int_{\mathcal{X}^m} \prod_{i=1}^m f(y_i) d\overset{\otimes m}{\nu}(y) = \left(\int_{\mathcal{X}} f(y_1) d\nu(y_1)\right)^m$, the above results show that

$$\int_{\mathcal{X}} f(x_1) d\mu(x_1) = \int_{\mathcal{X}} f(y_1) d\nu(y_1)$$

for any test function $f : \mathcal{X} \to \mathbb{R}$. It indicates that $\mu = \nu$. On the other hand, if $\mu = \nu$, we have $\overset{\otimes m}{\mu} = \overset{\otimes m}{\nu}$. We can construct the transportation plan $\bar{\gamma}(x, y) = \delta_x(y) \overset{\otimes m}{\mu}(x)$ for all $(x, y)$. Then, we find that

$$\mathcal{D}_d^m(\mu, \nu) \leq \mathbb{E}_{(X,Y) \sim \bar{\gamma}}[d(P_{X^m}, P_{Y^m})] = 0.$$

Therefore, we obtain the conclusion that $\mathcal{D}(\mu, \nu) = 0$ if and only if $\mu = \nu$.

Now, we demonstrate the triangle inequality property of $\mathcal{D}_d(., .)$, namely, we will show that for any probability measures $\mu_1, \mu_2$, and $\mu_3$, we have

$$\mathcal{D}_d^m(\mu_1, \mu_2) + \mathcal{D}_d^m(\mu_2, \mu_3) \geq \mathcal{D}_d^m(\mu_1, \mu_3).$$

The proof of this inequality is a direct application of gluing lemma (Berkes & Philipp, 1977; De Acosta, 1982). In particular, for any transportation plans $\gamma_1 \in \Pi(\overset{\otimes m}{\mu}_1, \overset{\otimes m}{\mu}_2)$ and $\gamma_2 \in \Pi(\overset{\otimes m}{\mu}_2, \overset{\otimes m}{\mu}_3)$, we can construct a probability measure $\xi$ on $\mathcal{X}^m \times \mathcal{X}^m \times \mathcal{X}^m$ such that $\xi(., ., \mathcal{X}^m) = \gamma_1(., .)$ and $\xi(\mathcal{X}^m, ., .) = \gamma_2(., .)$. Therefore, we find that

$$\mathbb{E}_{(X,Y) \sim \gamma_1}[d(P_{X^m}, P_{Y^m})] + \mathbb{E}_{(Y,Z) \sim \gamma_2}[d(P_{Y^m}, P_{Z^m})]$$
$$= \int_{(\mathcal{X}^m)^3} \left[d(P_{x^m}, P_{y^m}) + d(P_{y^m}, P_{z^m})\right] d\xi(x, y, z)$$
$$\geq \int_{(\mathcal{X}^m)^3} d(P_{x^m}, P_{z^m}) d\xi(x, y, z) \geq \mathcal{D}_d^m(\mu_1, \mu_3).$$

Taking the infimum on the LHS of the above inequality with respect to $\gamma_1$ and $\gamma_2$, we obtain the triangle inequality property of $\mathcal{D}(., .)$. As a consequence, we reach the conclusion of the theorem.

### A.2 PROOF OF THEOREM 2

To ease the presentation, for any product measures $\overset{\otimes m}{\mu}$ and $\overset{\otimes m}{\nu}$, we denote the following loss between $\overset{\otimes m}{\mu}$ and $\overset{\otimes m}{\nu}$ as follows:

$$\bar{\mathcal{D}}_d^m(\overset{\otimes m}{\mu}, \overset{\otimes m}{\nu}) := \inf_{\gamma \in \Pi(\overset{\otimes m}{\mu}, \overset{\otimes m}{\nu})} \mathbb{E}_{(X^m, Y^m) \sim \gamma}[d(P_{X^m}, P_{Y^m})].$$

From the definitions of the BoMb-OT losses, we have $\mathcal{D}_d^m(\mu, \nu) = \bar{\mathcal{D}}_d^m(\overset{\otimes m}{\mu}, \overset{\otimes m}{\nu})$ and $\widehat{\mathcal{D}}_d^{k,m}(\mu_n, \nu_n) = \bar{\mathcal{D}}_d^m(\overset{\otimes m}{\mu}_k, \overset{\otimes m}{\nu}_k)$. Using the similar proof argument as that of Theorem 1, we can check that $\bar{\mathcal{D}}_d^m$ satisfies the triangle inequality, namely, for any product measures $\overset{\otimes m}{\mu}, \overset{\otimes m}{\nu}$, and $\overset{\otimes m}{\eta}$, we have $\bar{\mathcal{D}}_d^m(\overset{\otimes m}{\mu}, \overset{\otimes m}{\nu}) + \bar{\mathcal{D}}_d^m(\overset{\otimes m}{\nu}, \overset{\otimes m}{\eta}) \geq \bar{\mathcal{D}}_d^m(\overset{\otimes m}{\mu}, \overset{\otimes m}{\eta})$.

From the above definition and properties of $\bar{\mathcal{D}}$, we find that

$$\left|\widehat{\mathcal{D}}_d^{k,m}(\mu_n, \nu_n) - \mathcal{D}_d^m(\mu, \nu)\right| = \left|\bar{\mathcal{D}}_d^m(\overset{\otimes m}{\mu}_k, \overset{\otimes m}{\nu}_k) - \bar{\mathcal{D}}_d^m(\overset{\otimes m}{\mu}, \overset{\otimes m}{\nu})\right|$$
$$\leq \left|\bar{\mathcal{D}}_d^m(\overset{\otimes m}{\mu}_k, \overset{\otimes m}{\nu}_k) - \bar{\mathcal{D}}_d^m(\overset{\otimes m}{\mu}, \overset{\otimes m}{\nu}_k)\right| + \left|\bar{\mathcal{D}}_d^m(\overset{\otimes m}{\mu}, \overset{\otimes m}{\nu}_k) - \bar{\mathcal{D}}_d^m(\overset{\otimes m}{\mu}, \overset{\otimes m}{\nu})\right|$$
$$\leq \bar{\mathcal{D}}_d^m(\overset{\otimes m}{\mu}_k, \overset{\otimes m}{\mu}) + \bar{\mathcal{D}}_d^m(\overset{\otimes m}{\nu}_k, \overset{\otimes m}{\nu}).$$

(i) Since $d \equiv W_p$ where $p \geq 1$, we have

$$d(P_{X^m}, P_{Y^m}) = W_p(P_{X^m}, P_{Y^m}) = \frac{1}{m^{1/p}} \left(\inf_\sigma \sum_{i=1}^m \|X_i - Y_{\sigma(i)}\|^p\right)^{1/p}$$
$$\leq \frac{1}{m^{1/p}} \sum_{i=1}^m \|X_i - Y_i\|,$$

where the infimum is taken over all possible permutations $\sigma$ of $\{1, 2, \ldots, m\}$. The above inequality indicates that

$$\bar{\mathcal{D}}_d^m(\overset{\otimes m}{\mu_k}, \overset{\otimes m}{\mu}) \leq \inf_{\gamma \in \Pi(\overset{\otimes m}{\mu_k}, \overset{\otimes m}{\mu})} \mathbb{E}_{(X^m, Y^m) \sim \gamma} \left[ \frac{1}{m^{1/p}} \sum_{i=1}^m \|X_i - Y_i\| \right] = m^{1-1/p} \cdot W_1(\mu_n, \mu).$$

Since $\mathcal{X}$ is a compact subset of $\mathbb{R}^N$, the results of (Dudley, 1969; Dobrić & Yukich, 1995; Fournier & Guillin, 2015) show that $W_1(\mu_n, \mu) = \mathcal{O}_P(n^{-1/N})$. Collecting these results together, we have

$$\bar{\mathcal{D}}_d^m(\overset{\otimes m}{\mu_k}, \overset{\otimes m}{\mu}) = \mathcal{O}_P\left( \frac{m^{1-1/p}}{n^{1/N}} \right).$$

With similar argument, we also find that $\bar{\mathcal{D}}_d^m(\overset{\otimes m}{\nu_k}, \overset{\otimes m}{\nu}) = \mathcal{O}_P\left( \frac{m^{1-1/p}}{n^{1/N}} \right)$. Putting all the above results together, we reach the conclusion of part (i) of the theorem.

(ii) The proof of part (ii) follows directly from the argument of part (i) and the results that $SW_p(\mu_n, \mu) = \mathcal{O}_P(n^{-1/2})$ and $SW_p(\nu_n, \nu) = \mathcal{O}_P(n^{-1/2})$ when $\mathcal{X}$ is a compact subset of $\mathbb{R}^N$ (Bobkov & Ledoux, 2019). Therefore, we obtain the conclusion of part (ii) of the theorem.

## B ADDITIONAL MATERIALS

**Computational complexity:** We now discuss the computational complexities of the BoMb-OT loss when $d \equiv \{W_p^\tau, SW_p\}$. When $d \equiv W_p^\tau$ for some given regularized parameter $\tau > 0$, we can use the Sinkhorn algorithm to compute $d(P_{X_i^m}, P_{Y_j^m})$ for any $1 \leq i, j \leq k$. The computational complexity of the Sinkhorn algorithm for computing it is of the order $\mathcal{O}(m^2)$. Therefore, the computational complexity of computing the cost matrix from the BoMb-OT is of the order $\mathcal{O}(m^2 k^2)$. Given the cost matrix, the BoMb-OT loss can be approximated with the complexity of the order $\mathcal{O}(k^2/\varepsilon^2)$ where $\varepsilon$ stands for the desired accuracy. As a consequence, the total computational complexity of computing the BoMb-OT is $\mathcal{O}(m^2 k^2 + k^2/\varepsilon^2)$.

When $d \equiv SW_p$, the computational complexity for computing $d(P_{X_i^m}, P_{Y_j^m})$ is of the order $\mathcal{O}(m \log m)$ for any $1 \leq i, j \leq k$. It shows that the complexity of computing the cost matrix from the BoMb-OT loss is of the order $\mathcal{O}(m(\log m)k^2)$. Given the cost matrix, the complexity of approximating the BoMb-OT is at the order of $\mathcal{O}(k^2/\varepsilon^2)$. Hence, the total complexity is at the order of $\mathcal{O}(m(\log m)k^2 + k^2/\varepsilon^2)$.

**Connection to hierarchical OT:** At the first sight, the BoMb-OT may look similar to hierarchical optimal transport (HOT). However, we would like to specify the main difference between the BoMb-OT and the HOT. In particular, on the one hand, the HOT comes from the hierarchical structure of data. For example, Yurochkin et al. (Yurochkin et al., 2019) consider optimal transport problems on both the document level and the word level for document representation. In the paper (Luo et al., 2020), HOT is proposed to handle multi-view data which is collected from different sources. Similarly, a hierarchical formulation of OT is proposed in (Lee et al., 2019b) to leverage cluster structure in data to improve alignment. On the other hand, the BoMb-OT makes *no assumption* about the hierarchical structure of data. We consider the optimal coupling of mini-batches as we want to improve the quality of mini-batch loss and its transportation plan.

**Entropic regularized population BoMb-OT:** We now consider an entropic regularization of the population BoMb-OT, which is particularly useful for reducing the computational complexity of the BoMb-OT. In particular, the *entropic regularized population BoMb-OT* (eBoMb-OT) between two probability measures $\mu$ and $\nu$ admits the following form:

$$\mathcal{E}\mathcal{D}_d^m(\mu, \nu) := \min_{\gamma \in \Pi(\overset{\otimes m}{\mu}, \overset{\otimes m}{\nu})} \mathbb{E}_{(X^m, Y^m) \sim \gamma}[d(P_{X^m}, P_{Y^m})] + \lambda \cdot \mathrm{KL}(\gamma | \overset{\otimes m}{\mu} \otimes \overset{\otimes m}{\nu}), \quad (4)$$

where $\lambda > 0$ stands for a chosen regularized parameter. From the above definition, the population eBoMb-OT is an interpolation between the population BoMb-OT and the population m-OT. On the one hand, when $\lambda \to 0$, we have $\mathcal{E}\mathcal{D}_d(\mu, \nu)$ converges to $\mathcal{D}_d(\mu, \nu)$. On the other hand, when $\lambda \to \infty$, the joint distribution $\gamma$ approaches $\overset{\otimes m}{\mu} \otimes \overset{\otimes m}{\nu}$ and $\mathcal{E}\mathcal{D}_d(\mu, \nu)$ converges to $U_d(\mu, \nu)$. The

transportation plan of the eBoMb-OT can be derived by simply replacing the coupling between mini-batches in Definition 3 by the coupling found by the Sinkhorn algorithm.

**Sparsity of transportation plan from BoMb-OT:** We assume that we are dealing with two empirical measures of $n$ supports. For $d = W_p$, the optimal transportation plan contains $n$ entries that are not zero. In the m-OT case, with $k$ mini-batches of size $m$, the m-OT's transportation plan contains at most $k^2 m$ non-zero entries. On the other hand, with $k$ mini-batches of size $m$, the BoMb-OT's transportation plan has at most $km$ non-zero entries. For $d = W_p^\tau$ ($\tau > 0$), the optimal transportation plans contains at most $n^2$ non-zero entries. In this case, the m-OT provides transportation plans that have at most $k^2 m^2$ non-zero entries. the BoMb-OT's transportation plans contain at most $km^2$ non-zero entries. The sparsity is useful in the re-forward step of the BoMb-OT algorithm since we can skip pair of mini-batches that has zero mass to save computation.

**Non-OT choice of $d$:** We would like to recall that $d$ in Definition 3 could be any discrepancy between empirical measures on mini-batches e.g. maximum mean discrepancy (MMD), etc. In this case, we can see the outer optimal transport between measures over mini-batches as an additional layer to incorporate the OT property into the final loss. However, it is not easy to define the notion of a transportation plan in these cases.

## C  APPLICATIONS AND BoMb-OT ALGORITHMS

In this section, we collect the details of applications that mini-batch optimal transport is used in practice including deep generative models, deep domain adaptation, color transfer, and approximate Bayesian computation. Moreover, we present corresponding algorithms of these applications with our new mini-batch scheme BoMb-OT.

### C.1  DEEP GENERATIVE MODEL

**Task description:** We first consider the applications of the m-OT and the BoMb-OT into parametric generative modeling. The goal of parametric generative modeling is to estimate a parameter of interest, says $\theta$, which belongs to a parameter space $\Theta$. Each value of $\theta$ induces a model distribution $\mu_\theta$ over the data space, and we want to find the optimal parameter $\theta^*$ which has $\mu_{\theta^*}$ as the closest distribution to the empirical data distribution $\nu_n$ under a discrepancy (e.g. Wasserstein distance). In deep learning setting, $\theta$ is the weight of a deep neural network that maps from a low dimensional manifold $Z$ to the data space $X$, and the model distribution $\mu_\theta$ is a push-forward distribution $G_\theta \sharp p(z)$ for $p(z)$ is a white noise distribution on $Z$ (e.g. $\mathcal{N}(0, I)$). By using mini-batching schemes, we can estimate this parameter by minimizing the BoMb-OT loss in Definition 3:

$$\theta^* \leftarrow \underset{\theta \in \Theta}{\arg\min} \, \hat{D}_d^{k,m}(\mu_{\theta,n}, \nu_n), \tag{5}$$

where $\mu_{\theta,n}$ is the empirical distribution of $\mu_\theta$. The mentioned optimization is used directly in learning generative model on MNIST dataset with $d \equiv (SW_2, W_2^\tau)$ in our experiments.

**Algorithms:** The algorithm for deep generative model with BoMb-OT is given in Algorithm 1. This algorithm is used to train directly the generative model on MNIST dataset.

**Metric learning:** Since $L_2$ distance is not a natural distance on the space of images such as CelebA, metric learning was introduced by (Genevay et al., 2018; Deshpande et al., 2018) as a key step in the application of generative models. The general idea is to learn a parametric ground metric cost $c_\phi$:

$$c_\phi(x, y) = \|f_\phi(x) - f_\phi(y)\|_2, \tag{6}$$

where $f_\phi : \mathcal{X} \to \mathbb{R}^h$ is a non-linear function that map from the data space to a feature space where $L_2$ distance is meaningful.

The methodology to learn the function $f_\phi$ depends on the choice of $d$. For example, when $d = W_2^\tau$, authors in (Genevay et al., 2018) seek for $\phi$ by solve following optimization:

$$\max_{f_\phi} \hat{D}_d^{k,m}(f_\phi \sharp \mu_{\theta,n}, f_\phi \sharp \nu_n), \tag{7}$$

---

**Algorithm 1** Mini-batch Deep Generative Model with BoMb-OT

---

**Input:** $k$, $m$, data $\mathcal{X}$, prior distribution $p(z)$, chosen mini-batch loss $L_{\text{DGM}} \in \{W_2, W_2^\tau, SW_2\}$
Initialize $G_\theta$ on GPU;
**while** $\theta$ does not converge **do**
    —— *On computer memory*——
    Sample indices uniformly $I_{X_1^m}, \ldots, I_{X_k^m}$ on $\mathcal{X}$
    Sample $Z_1^m, \ldots, Z_k^m$ from $p(z)$
    Intialize $C \in \mathbb{R}^{k \times k}$
    **for** $i = 1$ **to** $k$ **do**
        **for** $j = 1$ **to** $k$ **do**
            —— *On GPU, autograd off*——
            Load $X_i^m$ to GPU from $I_{X_i^m}$
            Load $Z_j^m$ to GPU
            Compute $Y_j^m \leftarrow G_\theta(Z_j^m)$
            Compute $C_{ij} \leftarrow L_{\text{DGM}}(P_{X_i^m}, P_{Y_j^m})$
        **end for**
    **end for**
    —— *On computer memory*——
    Solve $\pi \leftarrow \text{OT}(\boldsymbol{u}_k, \boldsymbol{u}_k, C)$ (entropic OT for eBoMb-OT)
    $\text{grad}_\theta \leftarrow 0$
    **for** $i = 1$ **to** $k$ **do**
        **for** $j = 1$ **to** $k$ **do**
            **if** $\pi_{ij} \neq 0$ **then**
                —— *On GPU, autograd on*——
                Load $X_i^m$ to GPU from $I_{X_i^m}$
                Load $Z_j^m$ to GPU
                Compute $Y_j^m \leftarrow G_\theta(Z_j^m)$
                Compute $C_{ij} \leftarrow L_{\text{DGM}}(P_{X_i^m}, P_{Y_j^m})$
                $\text{grad}_\theta \leftarrow \text{grad}_\theta + \pi_{ij}\nabla_\theta C_{ij}$
            **end if**
        **end for**
    **end for**
    —— *On GPU*——
    $\theta \leftarrow \text{Adam}(\theta, \text{grad}_\theta)$
**end while**

---

In (Deshpande et al., 2018), another metric learning technique is used for $d = SW_2$:

$$\max_{f_\phi} \max_{g_\psi} \mathbb{E}_{x \sim \mu_{\theta,n}, y \sim \nu_n}[\log g_\psi(f_\phi(x)) + \log(1 - g_\psi(f_\phi(y)))], \tag{8}$$

where $g_\psi : \mathbb{R}^h \to [0, 1]$.

Learning the metric or $f_\phi$ is carried out simultaneously with learning the generative model in practice, namely, after one gradient step for the parameter $\theta$, $\phi$ will be updated by one gradient step.

## C.2 DEEP DOMAIN ADAPTATION

We adapt the BoMb-OT into DeepJDOT (Bhushan Damodaran et al., 2018) which is a famous unsupervised domain adaptation method based on the m-OT. In particular, we aim to learn an embedding function $G_\theta : \mathcal{X} \to \mathcal{Z}$ which maps data to the latent space; and a classifier $F_\phi : \mathcal{Z} \to \mathcal{Y}$ which maps the latent space to the label space on the target domain. For a given number of the mini-batches $k$ and the size of mini-batches $m$, the goal is to minimize the following objective

---

**Algorithm 2** Mini-batch Deep Domain Adaptation with BoMb-OT

---

**Input:** $k$, $m$, source domain $(S, Y)$, target domain $T$, chosen cost $L_{\text{DA}}$ in Equation 10.
Initialize $G_\theta$ (parametrized by $\theta$), $F_\phi$ (parametrized by $\phi$)
**while** $(\theta, \phi)$ do not converge **do**
    —— *On computer memory*——
    Sample indices uniformly $I_{S_1^m, Y_1^m}, \ldots, I_{S_k^m, Y_k^m}$ on $(S, Y)$
    Sample indices uniformly $I_{T_1^m}, \ldots, I_{T_k^m}$ on $T$
    Initialize $C \in \mathbb{R}^{k \times k}$
    **for** $i = 1$ **to** $k$ **do**
        **for** $j = 1$ **to** $k$ **do**
            —— *On GPU, autograd off*——
            Load $(S_i^m, Y_i^m)$ to GPU from $I_{S_i^m, Y_i^m}$
            Load $T_j^m$ to GPU from $I_{T_j^m}$
            Compute $mC \leftarrow L_{\text{DA}}(S_i^m, Y_i^m, T_j^m, G_\theta, F_\phi)$ (Equation 10)
            $\boldsymbol{u}_m \leftarrow \left(\frac{1}{m}, \ldots, \frac{1}{m}\right)$
            Compute $C_{ij} \leftarrow \text{OT}(\boldsymbol{u}_m, \boldsymbol{u}_m, mC)$
        **end for**
    **end for**
    —— *On computer memory*——
    $\boldsymbol{u}_k \leftarrow \left(\frac{1}{k}, \ldots, \frac{1}{k}\right)$
    Solve $\gamma \leftarrow \text{OT}(\boldsymbol{u}_k, \boldsymbol{u}_k, C)$ (entropic OT for eBoMb-OT)
    $\text{grad}_\theta \leftarrow 0$
    $\text{grad}_\phi \leftarrow 0$
    **for** $i = 1$ **to** $k$ **do**
        —— *On GPU, autograd on*——
        Load $(S_i^m, Y_i^m)$ to GPU from $I_{S_i^m, Y_i^m}$
        $\text{grad}_\theta \leftarrow \text{grad}_\theta + \frac{1}{k}\frac{1}{m}\nabla_\theta L_s(Y_i^m, F_\phi(G_\theta(S_i^m)))$ (Equation 9)
        $\text{grad}_\phi \leftarrow \text{grad}_\phi + \frac{1}{k}\frac{1}{m}\nabla_\phi L_s(Y_i^m, F_\phi(G_\theta(S_i^m)))$ (Equation 9)
        **for** $j = 1$ **to** $k$ **do**
            **if** $\gamma_{ij} \neq 0$ **then**
                —— *On GPU, autograd on*——
                Load $T_j^m$ to GPU from $I_{T_j^m}$
                Compute $mC \leftarrow L_{\text{DA}}(S_i^m, Y_i^m, T_j^m, G_\theta, F_\phi)$ (Equation 10)
                Compute $C_{ij} \leftarrow \text{OT}(\boldsymbol{u}_m, \boldsymbol{u}_m, mC)$
                $\text{grad}_\theta \leftarrow \text{grad}_\theta + \gamma_{ij}\nabla_\theta C_{ij}$
                $\text{grad}_\phi \leftarrow \text{grad}_\phi + \gamma_{ij}\nabla_\phi C_{ij}$
            **end if**
        **end for**
    **end for**
    —— *On GPU*——
    $\theta \leftarrow \text{Adam}(\theta, \text{grad}_\theta)$
    $\phi \leftarrow \text{Adam}(\theta, \text{grad}_\phi)$
**end while**

---

function:

$$\min_{G_\theta, F_\phi} L_{\text{DA}} = \left[ \frac{1}{k}\frac{1}{m}\sum_{i=1}^{k}\sum_{j=1}^{m} L_s(y_{ij}, F_\phi(G_\theta(s_{ij}))) \right.$$

$$\left. + \min_{\gamma \in \Pi(\boldsymbol{u}_k, \boldsymbol{u}_k)}\sum_{i=1}^{k}\sum_{j=1}^{k}\gamma_{ij} \times \min_{\pi \in \Pi(\boldsymbol{u}_m, \boldsymbol{u}_m)} \langle C_{S_i^m, Y_i^m, T_j^m}^{G_\theta, F_\phi}, \pi \rangle \right], \tag{9}$$

where $L_s$ is the source loss function (e.g. classification loss), $S_1^m, \ldots, S_k^m$ are source mini-batches which are sampled with or without replacement from the source domain $S^m \in \mathcal{X}^m$, $Y_1^m, \ldots, Y_k^m$ are corresponding labels of $S_1^m, \ldots, S_k^m$, with $S_i^m = \{s_{i1}, \ldots, s_{im}\}$ and $Y_1^m := \{y_{i1}, \ldots, y_{im}\}$. Similarly, $T_1^m, \ldots, T_k^m$ ($T_1^m := \{t_{i1}, \ldots, t_{im}\}$) are target mini-batches which are sampled with or

without replacement from the target domain $\mathcal{T}^m \in \mathcal{X}^m$. The cost matrix $C_{S_i^m, Y_i^m, T_i^m}^{G,F}$ denoted $mC$ is defined as follows:

$$mC_{1 \leq z_1, z_2 \leq m} = \alpha||G(s_{iz_1}) - G(t_{jz_2})||^2 + \lambda_t L_t(y_{iz_1}, F(G(t_{jz_2}))), \quad (10)$$

where $L_t$ is the target loss function, $\alpha$ and $\lambda_t$ are hyper-parameters that control two terms.

## C.3 COLOR TRANSFER

In this appendix, we carry out more experiments on color transfer with various settings. In detail, we compare the m-OT and the BoMb-OT in different domains of images, and we demonstrate visually the interpolation property of the eBoMb-OT by varying the regularization parameter.

**Methodology:**    In our experiment, we first compress both the source image and the target image using K-means clustering with 3000 components. After that, we look for the transportation plans between cluster centers of two images from different approaches, the m-OT, the BoMb-OT, and the full-OT. Next, we change the values of all pixels in a cluster of the source image based on the value of the corresponding cluster centers of the target image with the previously found plans. We present the algorithm that we use in color transfer with the BoMb-OT in Algorithm 3. This algorithm is adapted from the algorithm that is used for the m-OT in (Fatras et al., 2020).

---

**Algorithm 3** Color Transfer with BoMb-OT

---

**Input:** $k, m, T$ source image $X_s \in \mathbb{R}^{n \times 3}$, target image $X_t \in \mathbb{R}^{n \times 3}$
Initialize $Y_s \in \mathbb{R}^{n \times 3}$
**for** $t = 1$ **to** $T$ **do**
    Initialize $C \in \mathbb{R}^{k \times k}$
    Sample indices $I_{X_1^m}, \ldots, I_{X_k^m}$ from $X_s$
    Sample indices $I_{Y_1^m}, \ldots, I_{Y_k^m}$ from $X_t$
    **for** $i = 1$ **to** $k$ **do**
        **for** $j = 1$ **to** $k$ **do**
            Load $X_i^m$ from $I_{X_i^m}$
            Load $Y_i^m$ from $I_{Y_i^m}$
            Compute cost matrix $M$ between $X_i^m$ and $Y_i^m$
            $\pi \leftarrow \arg\min_{\pi \in \Pi(\boldsymbol{u}_m, \boldsymbol{u}_m)} \langle M, \pi \rangle$
            $C_{ij} \leftarrow \langle M, \pi \rangle$
        **end for**
    **end for**
    $\gamma \leftarrow \arg\min_{\gamma \in \Pi(\boldsymbol{u}_k, \boldsymbol{u}_k)} \langle C, \gamma \rangle$
    **for** $i = 1$ **to** $k$ **do**
        **for** $j = 1$ **to** $k$ **do**
            **if** $\gamma_{ij} \neq 0$ **then**
                Load $X_i^m$ from $I_{X_i^m}$
                Load $Y_i^m$ from $I_{Y_i^m}$
                Compute cost matrix $M$ between $X_i^m$ and $Y_i^m$
                $\pi \leftarrow \arg\min_{\pi \in \Pi(\boldsymbol{u}_m, \boldsymbol{u}_m)} \langle M, \pi \rangle$
                $Y_s|_{I_{X_i^m}} \leftarrow m\gamma_{ij} \, \pi \cdot X_t|_{I_{Y_j^m}}$
            **end if**
        **end for**
    **end for**
**end for**
**Output:** $Y_s$

---

## C.4 APPROXIMATE BAYESIAN COMPUTATION (ABC)

In this appendix, details of Approximate Bayesian Computation (ABC) and the usage of mini-batches with BoMb-OT are discussed.

---

**Algorithm 4** Approximate Bayesian Computation

---

**Input:** Generative model $p(x, \theta)$, observation $\{x_i\}_{i=1}^n$, number of iterations $T$, discrepancy measure $D$, tolerance threshold $\epsilon$, number of particles $m$, and summary statistics $s$ (optional).
**for** $t = 1$ **to** $T$ **do**
   **repeat**
      Sample $\theta \sim p(\theta)$
      Sample $\{y_i\}_{i=1}^m \sim p(x|\theta)$
   **until** $D(s(\{y_i\}_{i=1}^m), s(\{x_i\}_{i=1}^n)) \leq \epsilon$
   $\theta_t = \theta$
**end for**
**Output:** $\{\theta_t\}_{t=1}^T$

---

**Review on ABC:** The central task of Bayesian inference is to estimate a distribution of parameter $\theta \in \Theta$ given $n$ data points $\{x_i\}_{i=1}^n$ and a generative model $p(x, \theta)$. Using Bayes rule, the posterior can be written as $p(\theta|x_{1:n}) := \frac{p(x_{1:n}|\theta)p(\theta)}{p(x_{1:n})}$. Generally, the posterior is intractable since we cannot evaluate the normalizing constant $p(x_{1:n})$ (or the evidence). It leads to the usage of approximate Bayesian Inference, e.g., Markov Chain Monte Carlo and Variational Inference. However, in some settings, the likelihood function $p(x_{1:n}|\theta)$ cannot be evaluated such as implicit generative models. In these cases, Approximate Bayesian Computation (or likelihood-free inference) is a good framework to infer the posterior distribution since it only requires the samples from the likelihood function.

We present the algorithm of ABC in Algorithm 4 that is used to obtain posterior samples of $\theta$. The thing that sets ABC apart is that it can be implemented in distributed ways, and its posterior can be shown to have the desirable theoretical property of converging to the true posterior when $\epsilon \to 0$. However, the performance of ABC depends on the choice of the summary statistics $s$ (e.g., empirical mean and empirical variance) and the discrepancy $D$. In practice, constructing sufficient statistics is not easy. Thus, a discrepancy between empirical distributions is used to avoid this non-trivial task. Currently, Wasserstein distances have drawn a lot of attention from ABC's community because they provide a meaningful comparison between non-overlap probability distributions (Bernton et al., 2019a; Nadjahi et al., 2020). When the number of particles $m$ in Algorithm 4 is the largest OT problem that can be solved by the current computational resources, the mini-batch approach can be utilized to obtain more information about two measures, namely, more supports in sample space can be used. In particular, we utilize the mini-batch OT losses as the discrepancy $D$ in Algorithm 4. The detail of the application of BoMb-OT in ABC is given in Algorithm 5.

# D  ADDITIONAL EXPERIMENTS

In this appendix, we provide additional experimental results that are not shown in the main paper. In Appendix D.1, we present random generated images on MNIST dataset, CIFAR10 dataset, CelebA dataset, and training time comparison between the m-OT and the BoMb-OT. Furthermore, we give detailed result on deep domain adaptation in Appendix D.2 and we also provide the computational time. In Appendix D.3 we carry out more experiments on color transformation with various settings on different images and their corresponding color palettes. Moreover, details of setting of the ABC experiment and additional comparative experiments are presented in Appendix D.4. After that, we compare the m-OT and the BoMb-OTs on gradient flow application in Appendix D.5. Finally, we investigate the behavior of the m-OT and the (e)BoMb-OT in estimating the optimal transportation plan in Appendix D.6.

## D.1  DEEP GENERATIVE MODEL

**Generated images:** We show the generated images on MNIST dataset in Figure 4, the generated images on CelebA in Figure 5, the generated images on CIFAR10 in Figure 6. For both choices of $d$ ($SW_2$, $W_2^\tau$), we can see that the images from (e)BoMb-OT is more realistic than the images from m-OT. This qualitative result supports the quantitative in Table 1 in the main text.

---

**Algorithm 5** Approximate Bayesian Computation with BoMb-OT

---

**Input:** Generative model $p(x, \theta)$, observation $\{x_i\}_{i=1}^n$, number of iterations $T$, optimal transport discrepancy measure $D_{\text{OT}}$, tolerance threshold $\epsilon$, number of particles $m$, number of mini-batches $k$, and summary statistics $s$ (optional).

**for** $t = 1$ **to** $T$ **do**
   **repeat**
      Sample $\theta \sim p(\theta)$
      Sample indices $I_{X_1^m}, \ldots, I_{X_k^m}$ from $\{x_i\}_{i=1}^n$
      Sample $\{y_i\}_{i=1}^n \sim p(x|\theta)$
      Sample indices $I_{Y_1^m}, \ldots, I_{Y_k^m}$ from $\{y_i\}_{i=1}^n$
      Initialize $C \in \mathbb{R}^{k \times k}$
      **for** $i = 1$ **to** $k$ **do**
         **for** $j = 1$ **to** $k$ **do**
            Load $X_i^m, Y_j^m$ from their indices $I_{X_i^m}, I_{Y_j^m}$
            $C_{ij} \leftarrow D_{\text{OT}}(X_i^m, Y_j^m)$
         **end for**
      **end for**
      $\boldsymbol{u}_k \leftarrow \left(\frac{1}{k}, \ldots, \frac{1}{k}\right)$
      $D \leftarrow \text{OT}(\boldsymbol{u}_k, \boldsymbol{u}_k, C)$
   **until** $D \leq \epsilon$
   $\theta_t = \theta$
**end for**
**Output:** $\{\theta_t\}_{t=1}^T$

---

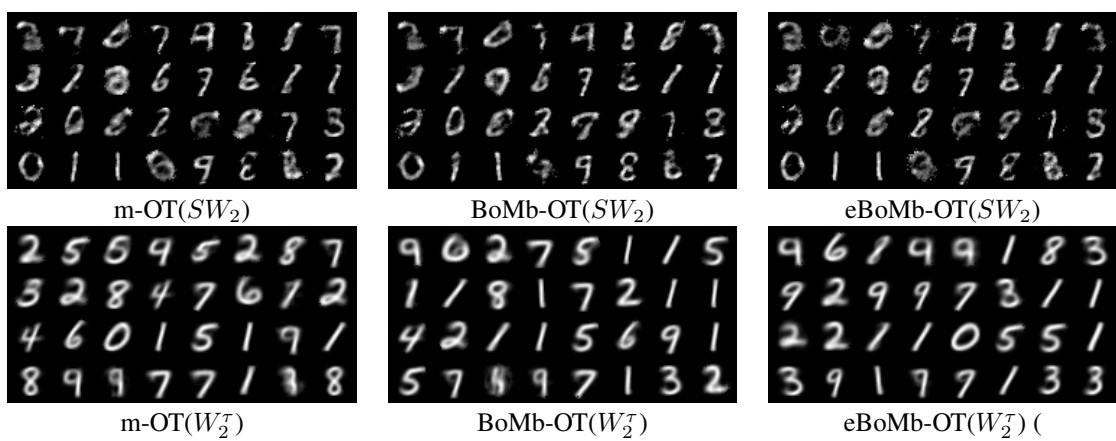

Figure 4: MNIST generated images from the m-OT and the (e)BoMb-OT for $(k, m) = (4, 100)$.

**Computational speed:** Table 3 details the computational speed of deep generative model when using mini-batch size $m = 100$. In general, the more the number of mini-batches is, the more complex the problem is. Increasing the number of mini-batches $k$ does decrease the number of iterations per second in all experiments. For both datasets, using the sliced Wasserstein distance $(d = SW_2)$ leads to higher iterations per second compared to the entropic Wasserstein distance $(d = W_2^\tau)$ on the high dimensional space. This result is expected since the slicing approach aims to reduce the computational complexity. On the MNIST dataset, we observe that the entropic regularized version of the BoMb-OT (eBoMb-OT) is the slowest for both choices of $d$. The m-OT is the fastest approach with the number of iterations per second of $7.89, 2.54$, and $0.85$ for $k = 2, 4$, and $8$ respectively. In contrast, the BoMb-OT is faster than the m-OT for CIFAR10 and CelebA dataset because we set a high value for the regularized parameter (e.g. $\tau = 50, 40$). For the CelebA dataset, although the eBoMb-OT is the slowest method when $d = SW_2$, it becomes the fastest approach if we set $d$ to $W_2^\tau$. The reason for this phenomenon is because of the sparsity of the (e)BoMb-OT's transportation plan between measures over mini-batches. In particular, pairs of mini-batches that are zero can be skipped

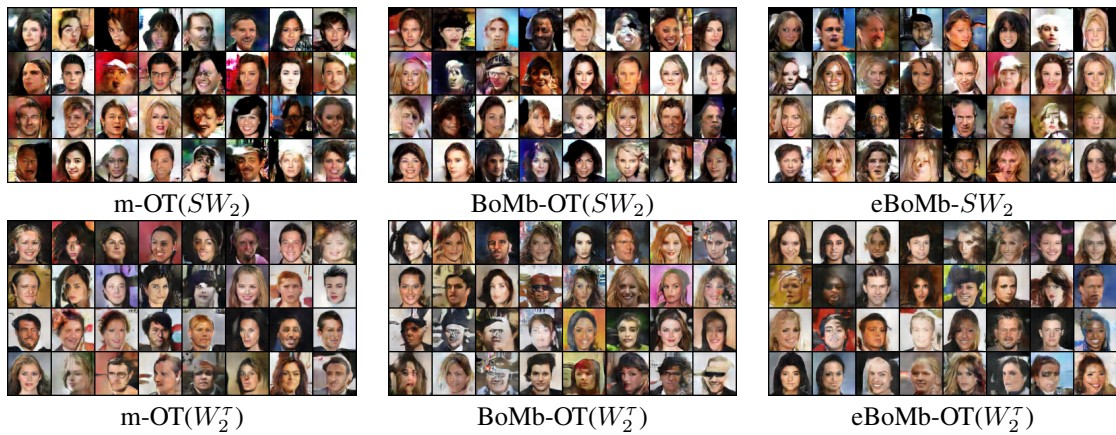

| m-OT($SW_2$) | BoMb-OT($SW_2$) | eBoMb-$SW_2$ |
| --- | --- | --- |
| m-OT($W_2^\tau$) | BoMb-OT($W_2^\tau$) | eBoMb-OT($W_2^\tau$) |

Figure 5: CelebA generated images from the m-OT and the (e)BoMb-OT for (k,m)=(4,100).

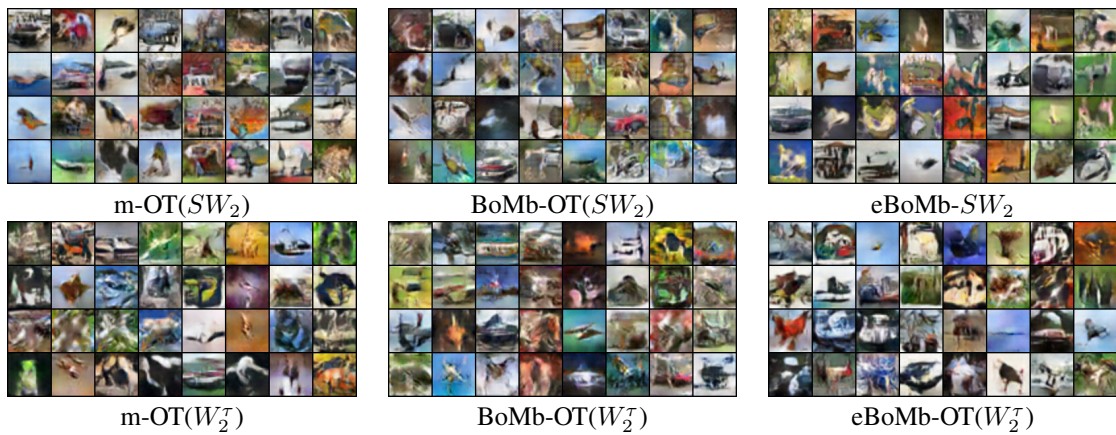

| m-OT($SW_2$) | BoMb-OT($SW_2$) | eBoMb-$SW_2$ |
| --- | --- | --- |
| m-OT($W_2^\tau$) | BoMb-OT($W_2^\tau$) | eBoMb-OT($W_2^\tau$) |

Figure 6: CIFAR10 generated images from the m-OT and the (e)BoMb-OT for (k,m)=(4,100).

Table 3: Number of iterations per second of deep generative models

| Dataset | $k$ | m-OT($W_2^\tau$) | BoMb-OT($W_2^\tau$) | eBoMb-OT ($W_2^\tau$) | m-OT($SW_2$) | BoMb-OT($SW_2$) | eBoMb-OT ($SW_2$) |
| --- | --- | --- | --- | --- | --- | --- | --- |
| MNIST | 1 | 27.27 | 27.27 | 27.27 | 54.55 | 54.55 | 54.55 |
| | 2 | **7.89** | 5.88 | 4.62 | **18.75** | 15.00 | 11.54 |
| | 4 | **2.54** | 2.11 | 1.36 | **6.00** | 5.36 | 3.41 |
| | 8 | **0.85** | 0.79 | 0.44 | **1.70** | 1.74 | 0.95 |
| CIFAR10 | 1 | 22.73 | 22.73 | 22.73 | 25.00 | 25.00 | 25.00 |
| | 2 | 6.94 | 5.81 | **7.35** | 9.62 | **10.00** | 7.58 |
| | 4 | 2.05 | **2.31** | 2.16 | 3.21 | **4.17** | 2.40 |
| | 8 | 0.53 | **0.78** | 0.59 | 0.97 | **1.54** | 0.71 |
| CelebA | 1 | 6.78 | 6.78 | 6.78 | 9.93 | 9.93 | 9.93 |
| | 2 | 1.92 | 1.75 | **2.52** | 3.38 | **3.55** | 2.65 |
| | 4 | 0.45 | 0.54 | **0.68** | 1.03 | **1.40** | 0.78 |
| | 8 | 0.11 | 0.15 | **0.18** | 0.29 | **0.51** | 0.22 |

(see in Algorithm 1. So, the (e)BoMb-OT can avoid estimating gradient of the neural networks of a bad pair of mini-batches.

Table 4: Effect of changing the mini-batch size on deep domain adaptation

| Scenario | $m$ | $k$ | Number of epochs | m-OT | BoMb-OT | Improvement |
|---|---|---|---|---|---|---|
| SVHN to MNIST | 10 | 32 | 64 | 69.41 | **88.49** | +19.08 |
| | 20 | 32 | 128 | 89.69 | **93.54** | +3.85 |
| | 50 | 32 | 320 | 93.09 | **95.59** | +2.40 |
| USPS to MNIST | 5 | 32 | 50 | 73.92 | **93.15** | +19.23 |
| | 10 | 32 | 100 | 92.96 | **95.06** | +2.10 |
| | 20 | 32 | 200 | 95.85 | **96.15** | +0.30 |

Table 5: Comparison between the two mini-batch schemes on deep domain adaptation on Office-Home datasets.

| $k$ | Methods | A2C | A2P | A2R | C2A | C2P | C2R | P2A | P2C | P2R | R2A | R2C | R2P | Avg |
|---|---|---|---|---|---|---|---|---|---|---|---|---|---|---|
| 2 | m-UOT | **56.22** | 75.04 | **80.51** | 64.81 | 74.50 | 75.08 | 65.88 | 52.23 | **80.01** | **74.21** | 60.00 | 82.97 | 70.15 |
| | eBoMb-UOT | 55.90 | **75.13** | **80.51** | **66.01** | **74.52** | **75.63** | **65.97** | **53.22** | 79.99 | 74.04 | **60.09** | **83.29** | **70.39** |
| 4 | m-UOT | 55.42 | 75.13 | 80.45 | **65.88** | 73.89 | 74.50 | 65.76 | 52.90 | 79.96 | 74.21 | 59.84 | 83.17 | 70.09 |
| | eBoMb-UOT | **56.06** | **75.15** | **80.63** | **65.88** | **73.91** | **75.30** | **65.84** | **53.24** | **80.17** | **74.29** | **60.18** | **83.31** | **70.33** |
| 8 | m-UOT | 56.04 | 75.08 | 80.45 | 65.76 | 73.40 | 74.68 | 65.76 | 53.04 | 79.92 | 74.29 | **60.14** | **83.24** | 70.15 |
| | eBoMb-UOT | **56.22** | **75.56** | **80.51** | **65.84** | **74.30** | **74.89** | **66.05** | **53.08** | **80.03** | **74.41** | 60.12 | 83.13 | **70.35** |

Table 6: Computational speed of deep DA when changing $k$

| Scenario | k | m-OT | BoMb-OT |
|---|---|---|---|
| SVHN to MNIST | 8 | 2.26 | **3.05** |
| | 16 | 0.60 | **1.06** |
| | 32 | 0.15 | **0.32** |
| USPS to MNIST | 8 | 6.00 | **9.00** |
| | 16 | 1.80 | **2.57** |
| | 32 | 0.45 | **0.75** |

Table 7: Computational speed of deep DA when changing $m$

| Scenario | m | m-OT | BoMb-OT |
|---|---|---|---|
| SVHN to MNIST | 10 | 0.33 | **0.61** |
| | 20 | 0.33 | **0.60** |
| | 50 | 0.15 | **0.32** |
| USPS to MNIST | 5 | 0.49 | **1.00** |
| | 10 | 0.49 | **1.00** |
| | 20 | 0.46 | **1.00** |

## D.2 DEEP DOMAIN ADAPTATION

In this section, we compare the performance of two mini-batch schemes on digits and Office-Home datasets.

**Comparison between the m-OT and BoMb-OT on digits datasets:** As seen in Table 4, the BoMb-OT produces better classification accuracy than the m-OT in all experiments. In addition, it leads to a huge performance improvement (over $19\%$) in comparison with the m-OT when the mini-batch size is small ($k = 10$ for SVHN to MNIST and $k = 5$ for USPS to MNIST). When the mini-batch size becomes larger, the performance of both methods also increases. An explanation for such outcome is that a large mini-batch size makes mini-batch loss approach to its population version.

**Comparison between the m-UOT and eBoMb-UOT on Office-Home dataset:** Table 5 illustrates the performance of m-UOT and eBoMb-UOT on Office-Home dataset when changing the number of mini-batches $k$ from 2 to 8. When $k = 4$, eBoMb-UOT achieves the classification accuracy higher than m-UOT on 11 out of 12 scenarios, resulting in an improvement of 0.24 on average.

**Computational speed:** The number of iterations per second of deep DA can be found in Table 6-7. Similar to deep generative model, we observe a phenomenon that the speed of both the m-OT and BoMb-OT decreases as $k$ increases. Interestingly, increasing the mini-batch size $m$ when adapting from USPS to MNIST barely affects the running speed of both methods. The run time of the m-OT nearly doubles that of the BoMb-OT. Specifically, the BoMb-OT averagely runs 1 iteration in a second while an iteration of the m-OT consumes roughly 2 seconds. Although having comparable time complexity, the BoMb-OT in practice runs faster than the m-OT in all deep DA experiments. This is because of the sparsity of the BoMb-OT's transportation plan between measures over mini-batches. In particular, pairs of mini-batches that are zero can be skipped (see in Algorithm 2). We would like to recall that, gradient estimation is the most time-consuming task in deep learning applications.

### D.3 COLOR TRANSFER

In this appendix, we compare the m-OT and the BoMb-OT in different domains of images including natural images and arts.

**Comparison between the m-OT and the BoMb-OT:** Next, we illustrate the color-transferred images using the m-OT and the BoMb-OT with two values of the number of mini-batches and the size of mini-batches $(k, m) = (10, 10)$ and $(k, m) = (20, 20)$. The number of incremental step $T$ (see in Algorithm 3) is set to 5000. We show the source images, target images, and corresponding transferred images in Figure 7-Figure 10. It is easy to see that the transferred images from the BoMb-OT look more realistic than the m-OT and the color is more similar to the target images. The color palette of transferred images also reinforces the above claim when it is closer to the color palette of the target images. According to those figures, increasing the number of mini-batches and the mini-batch size improves the results considerably.

**Computational speed:** For $k = 10, m = 10$, the m-OT has the speed of 103 iterations per second while the BoMb-OT has the speed of 100 iterations per second. For $k = 20, m = 20$, the speed of m-OT is about 20 iterations per second and the speed of the BoMb-OT is also about 20 iterations per second. It means that the additional $k \times k$ OT is not too expensive while it can improve the color palette considerably.

### D.4 APPROXIMATE BAYESIAN COMPUTATION (ABC)

**Settings:** We use the same setup as in (Nadjahi et al., 2020): there are $n = 100$ observations $\{x_i\}_{i=1}^n$ i.i.d from multivariate Gaussian $\mathcal{N}(\mu_*, \sigma_*^2 I_N)$ where $N$ is the dimension, $\mu_* \sim \mathcal{N}(0, I_N)$ and $\sigma_*^2 = 4$. The task is to estimate the posterior of $\sigma^2$ under the imaginary assumption that $\sigma^2$ follows inverse gamma distribution $\mathcal{IG}(1, 1)$. Under these assumptions, the posterior of $\sigma^2$ has the form $\mathcal{IG}(1 + n\frac{d}{2}, 1 + \frac{1}{2}\sum_{i=1}^n ||x_i - \nu_*||^2)$. For ABC, we use the m-OT and the BoMb-OT with the Wasserstein-2 ground metric for the acceptance-rejection sampling's criteria in sequential Monte Carlo ABC (Toni et al., 2009) (using implementation from pyABC (Klinger et al., 2018) with 100 particles and 10 iterations). The mini-batches' size is set in $\{8, 16, 32\}$ and the number of mini-batches is in $\{2, 4, 6, 8\}$.

**Results on ABC with mini-batches:** After obtaining all samples, we estimate their densities by utilizing Gaussian kernel density estimation and then plot the approximated posteriors and the true posterior in Figure 11. Here we run the algorithm with several values of $(k, m)$, namely, $m \in \{8, 16, 32\}$ and $k \in \{2, 4, 6, 8\}$. From these results, we see that a bigger $m$ usually returns a better posterior in both the m-OT and the BoMb-OT cases. Similarly, increasing $k$ also improves the performance of the two methods. Moreover, these graphs strengthen the claim that the BoMb-OT outperforms the m-OT in every setting of $(k, m)$ since its posteriors are always closer to the true posterior than those of the m-OT in both visual result and Wasserstein distance.

### D.5 NON-PARAMETRIC GENERATIVE MODEL VIA GRADIENT FLOW

In this appendix, we show the experiment that uses the m-OT and the (e)BoMb-OT in the gradient flow application.

**Task description:** Gradient flow is a non-parametric method to learn a generative model. Like every generative model, the goal of gradient flow is to mimic the data distribution $\nu$ by a distribution $\mu$. It leads to the functional optimization problem:

$$\min_\mu D(\mu, \nu), \tag{11}$$

where $D$ is a predefined discrepancy between two probability measures. So, a gradient flows can be constructed:

$$\partial_t \mu_t = -\nabla_{\mu_t} D(\mu_t, \nu) \tag{12}$$

We follow the Euler scheme to solve this equation as in (Feydy et al., 2019), starting from an initial distribution at time $t = 0$. In this paper, we choose $D$ be (e)BoMb-$W_2$ and m-$W_2$ for the sake of comparison between them.

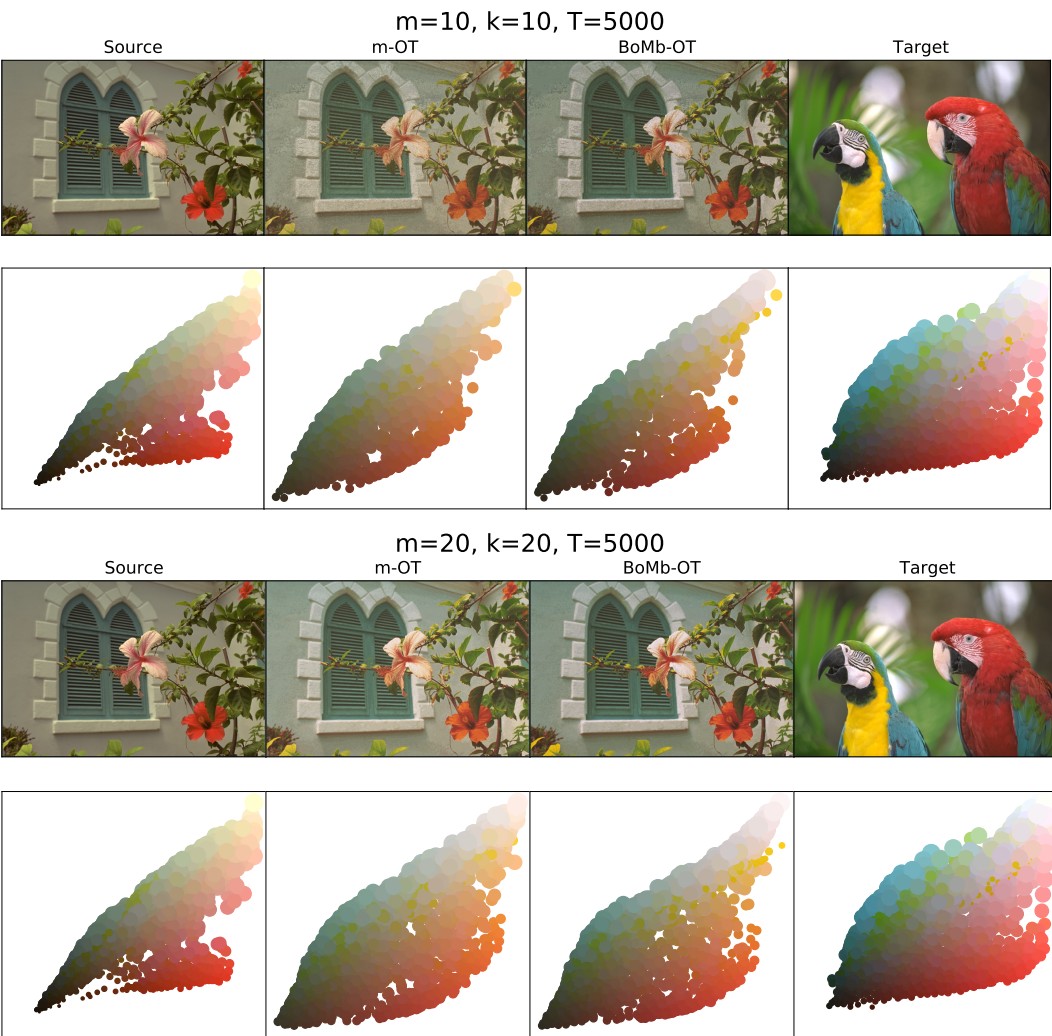

Figure 7: Experimental results on color transfer for the m-OT, the BoMb-OT on natural images with $(k, m) = (10, 10)$, $(k, m) = (20, 20)$, and $T = 5000$. Color palettes are shown under corresponding images.

We first consider the toy example as in (Feydy et al., 2019) and present our results in Figure 12. The task is to move the colorful empirical measure to the "S-shape" measure. Each measure has 1000 support points. Here, we choose $(k, m) = (4, 16)$, the OT loss is Wasserstein-2, and we use the Wasserstein-2 score to evaluate the performance of the mini-batching scheme. From Figure 12, BoMb-OT and eBoMb-OT provide better flows than m-OT, namely, Wasserstein-2 scores of BoMB-OT and eBoMb-OT are always lower than those of m-OT in every step. In addition, we do an extra setup with a higher of mini-batches, $(k, m) = (16, 16)$ to show the increasing the number of mini-batches improve the performance of both m-OT and (e)BoMb-OT. The result is shown in Figure 13. In this setting, the BoMb-OT still shows its favorable performance compared to the m-OT, namely, its Wasserstein-2 scores are still lower than the m-OT in every step.

**CelebA:** Let $\mu$ and $\nu$ denote the empirical measures defined over 5000 female images and 5000 male images in the CelebA dataset. Thus, we can present the transformation from a male face to be a female one by creating a flow from $\mu$ to $\nu$. Our setting experiment is the same as (Fatras et al., 2020). We first train an autoencoder on CelebA, then we compress two original measures to the low-dimensional measures on the latent space of the autoencoder. After having the latent measures, we run the Euler scheme to get the transformed measure then we decode it back to the data space by the autoencoder. The result is shown in Figure 14, we also show the closest female image (in sense of

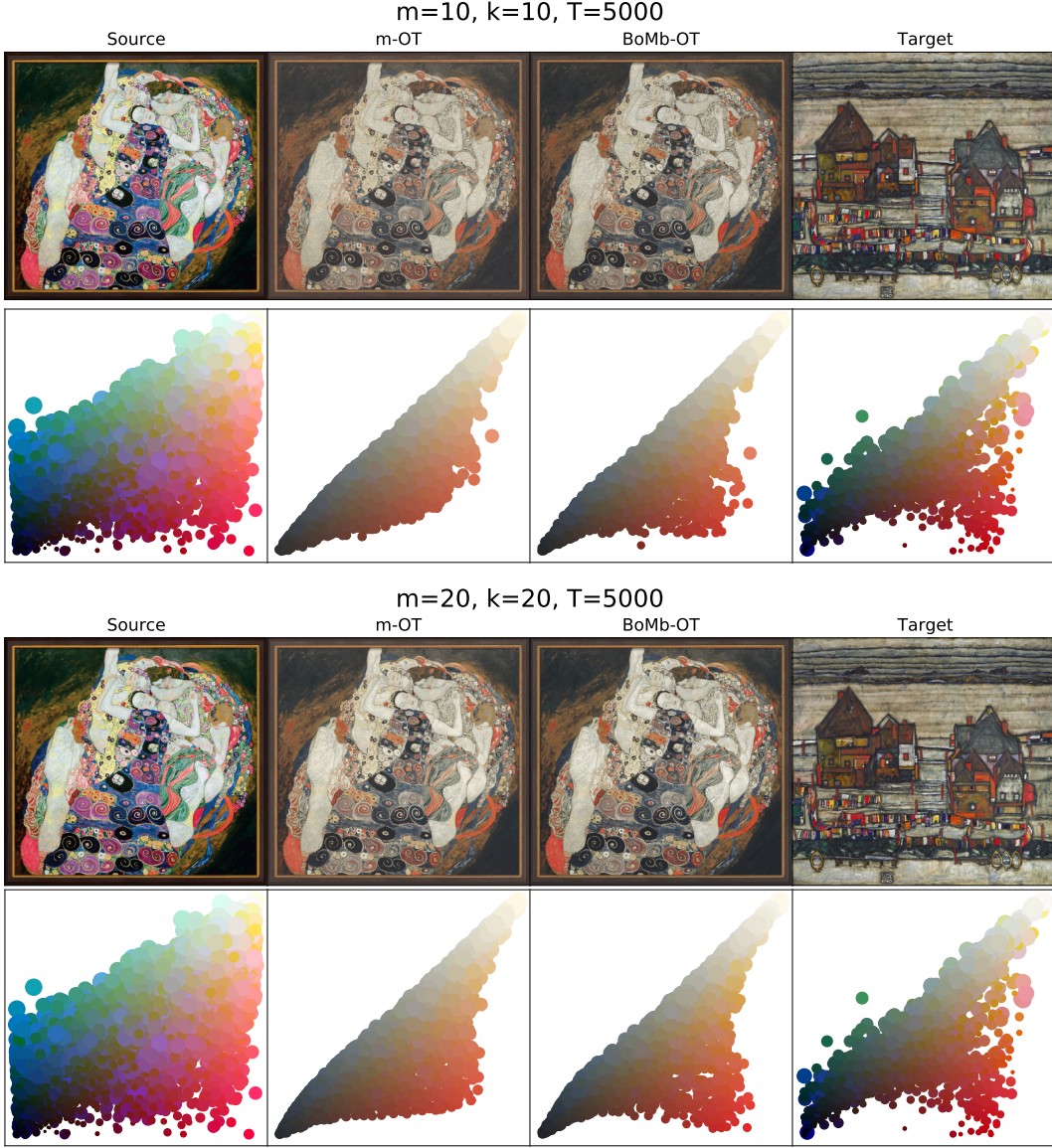

Figure 8: Experimental results on color transfer for the m-OT, the BoMb-OT on arts with $(k, m) = (10, 10)$, $(k, m) = (20, 20)$, and $T = 5000$. Color palettes are shown under corresponding images.

$L_2$ distance) to the final found image in each method and the corresponding $L_2$ distance between the middle step images and the nearest image. As shown, (e)BoMb-OT provides a better flow than m-OT does.

### D.6 TRANSPORTATION PLANS

In this appendix, we investigate deeper the behavior of the m-OT and the (e)BoMb-OT in estimating the transportation plan. We present a toy example to illustrate the transportation plans of the m-OT, the BoMb-OT, and the original OT (full-OT). In particular, we sample two empirical distributions $\mu_n$ and $\nu_n$, where $n = 10$, from $\mathcal{N}\left(\begin{bmatrix} 0 \\ 0 \end{bmatrix}, \begin{bmatrix} 1 & 0 \\ 0 & 1 \end{bmatrix}\right)$ and $\mathcal{N}\left(\begin{bmatrix} 4 \\ 4 \end{bmatrix}, \begin{bmatrix} 1 & -0.8 \\ -0.8 & 1 \end{bmatrix}\right)$ respectively.

**Transportation plans:** We plot the graph of samples matching and transportation matrices in Figure 15. When the size of mini-batches equals 2 and the number of mini-batches is set to be 20, the

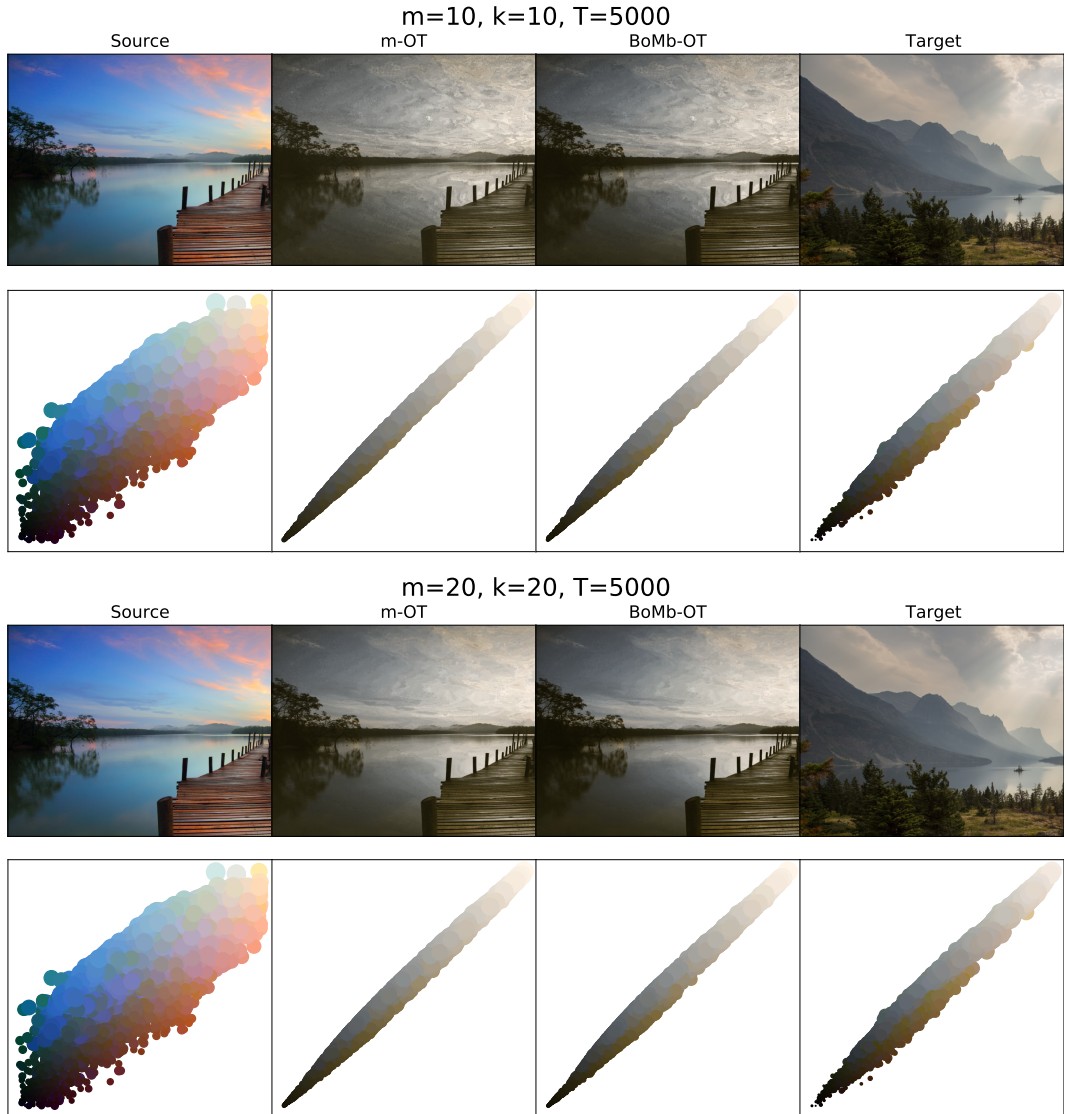

Figure 9: Experimental results on color transfer for the m-OT, the BoMb-OT on natural images with $(k, m) = (10, 10)$, $(k, m) = (20, 20)$, and $T = 5000$. Color palettes are shown under corresponding images.

m-OT approach produces messy OT matrices and disordered matching, meanwhile, the BoMb-OT can still concentrate the mass to meaningful entries of the transportation matrix, thus, its matching is acceptable. Increasing the size of mini-batches to 8, m-OT's performance improves significantly however its matching and its matrices are visibly still not good solutions. In contrast, the BoMb-OT is able to generate nearly optimal matchings and transportation matrices. Next, we test the m-OT and the BoMb-OT in real applications in the case of an extremely small number of mini-batches. When there are 8 mini-batches of size 2, the m-OT still performs poorly while the BoMb-OT creates a sparser transportation matrix that is closer to the optimal solution obtained by the full-OT. Similarly, with 2 mini-batches of size 8, the BoMb-OT has only 4 wrong matchings while that number of the m-OT is 10. In conclusion, the BoMb-OT is the better version for mini-batching with any value of the number of mini-batches and mini-batches size.

**Entropic transportation plan of the BoMb-OT:** We empirically show that the transportation plan of the eBoMb-OT, when the entropic regularization is sufficiently large, reverts into the m-OT's plan. The result is given in Figure 16. When $\lambda = 10^3$, the transportation plans of the eBoMb-OT

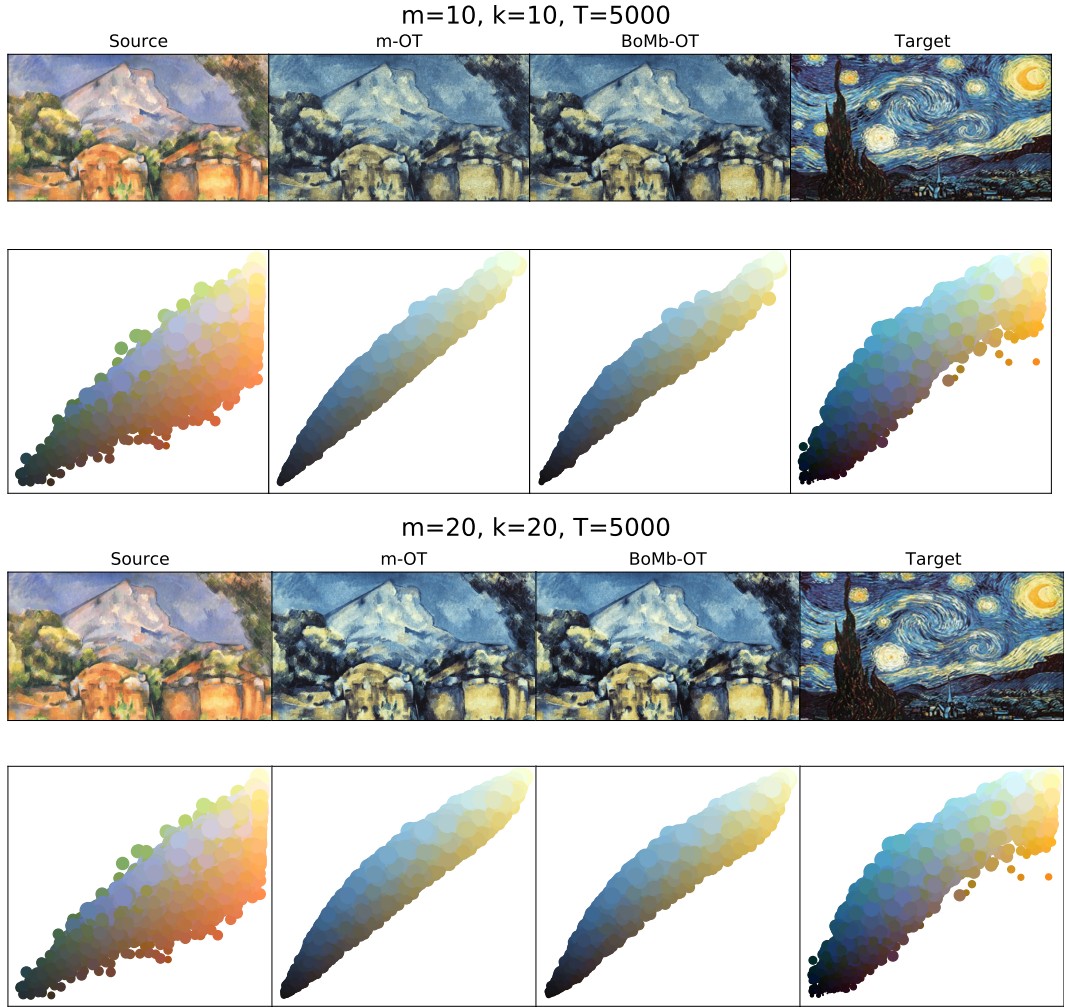

Figure 10: Experimental results on color transfer for the m-OT, the BoMb-OT on arts with $(k, m) = (10, 10)$, $(k, m) = (20, 20)$, and $T = 5000$. Color palettes are shown under corresponding images.

are identical to plans of the m-OT with every choice of $(k, m)$. However, when $\lambda = 0.1$, despite not being identical to full-OT, the plans produced by the eBoMb-OT are still close to the true optimal plan.

**Comparison with stochastic averaged gradient (SAG):** We compare the BoMb-OT with the stochastic averaged gradient (SAG) (Aude et al., 2016) for computing OT. The transportation matrices are given in Figure 17. We would like to recall that SAG still need to store full cost matrix ($n \times$) while the BoMb-OT only need to store smaller cost matrices ($m \times m$). We can see that the BoMb-OT gives better than transportation plan than SAG when $m = 2$. When $m = 8$, the BoMb-OT still seems to be better.

# E    EXPERIMENT SETTINGS

In this appendix, we collect some necessary experimental setups in the paper including generative model, gradient flow. We use POT (Flamary et al., 2021) for OT solvers, and the pyABC (Klinger et al., 2018) for the ABC experiments.

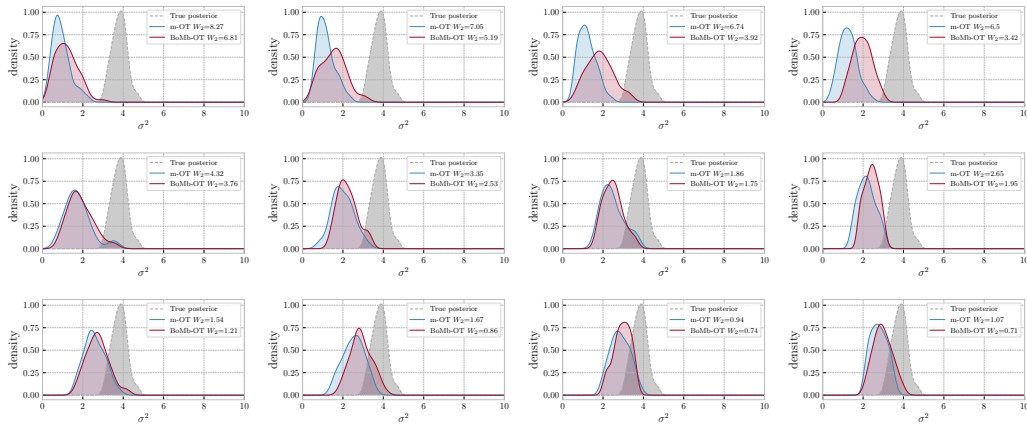

Figure 11: Approximated posteriors from ABC with m-OT and BoMb-OT. The first row, the second row, and the last row have $m = 8$, $m = 16$, and $m = 32$ respectively. In each row, the number of mini-batches $k$ are $k = 2, 4, 6$ and $k = 8$ from left to right.

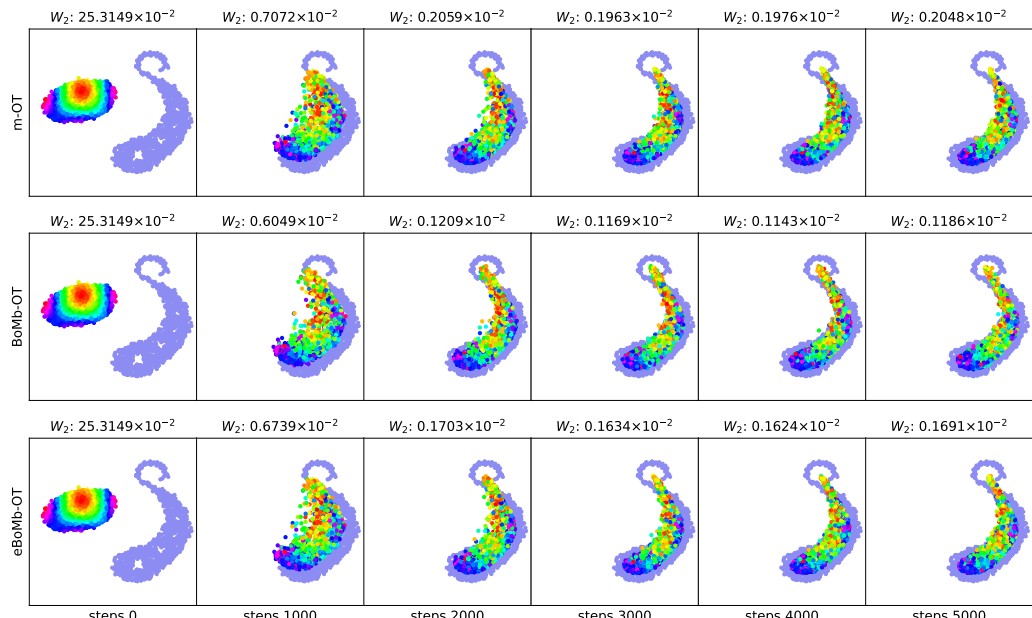

Figure 12: Comparison between (e)BoMb-OT and m-OT in gradient flow with $n = 1000$, $k = 4$, $m = 16$. The entropic regularized parameter of eBoMb-OT $\lambda$ is set to 0.01.

### E.1  DEEP GENERATIVE MODEL

**Wasserstein-2 scores:**  We use empirical distribution with 11000 samples that are obtained by sampling from the generative model and the empirical test set distribution respectively, then we compute discrete Wasserstein-2 distance via linear programming.

**FID scores:**  We use 10000 samples from the generative model and all test set images to compute FID score (Heusel et al., 2017).

**Parameter settings:**  We chose a learning rate equal to 0.0005, batch size equal to 100, number of epochs in MNIST equal to 100, number of epochs in CelebA equal to 25, number of epochs in CIFAR10 equal to 50. For $d = SW_2$ we use the number of projections $L = 1000$ on MNIST and $L = 100$ on CIFAR10 and CelebA. For $d = W_2^\tau$, we use $\tau = 1$ for MNIST,

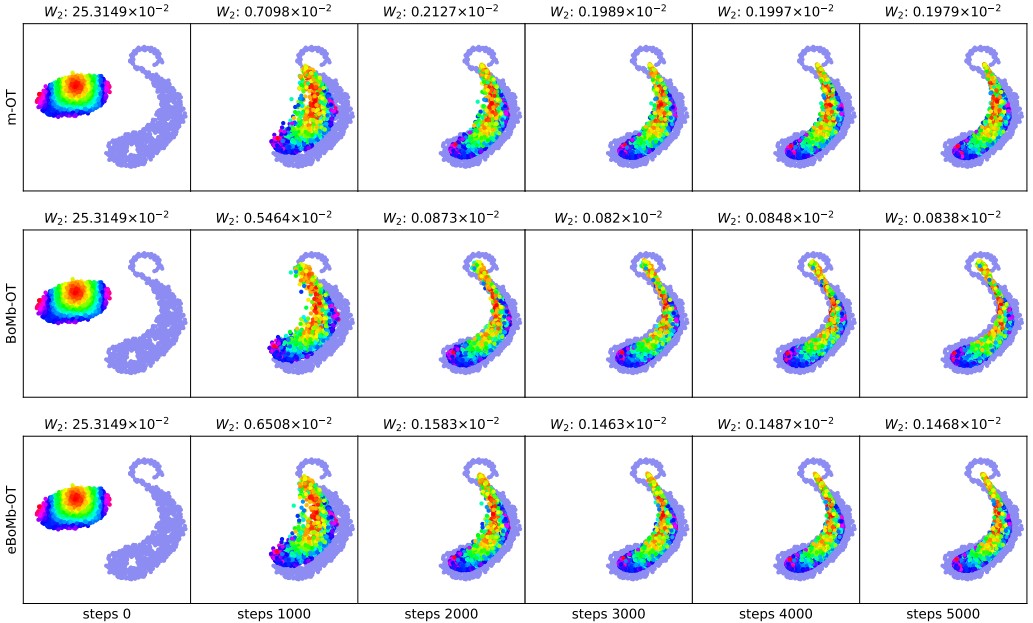

Figure 13: Visualization of the gradient flow provided by m-OT, BoMb-OT and eBoMb-OT with corresponding Wasserstein-2 scores ($n = 1000$, $(k, m) = (16, 16)$). The entropic regularized parameter of eBoMb-OT $\lambda$ is set to 0.01.

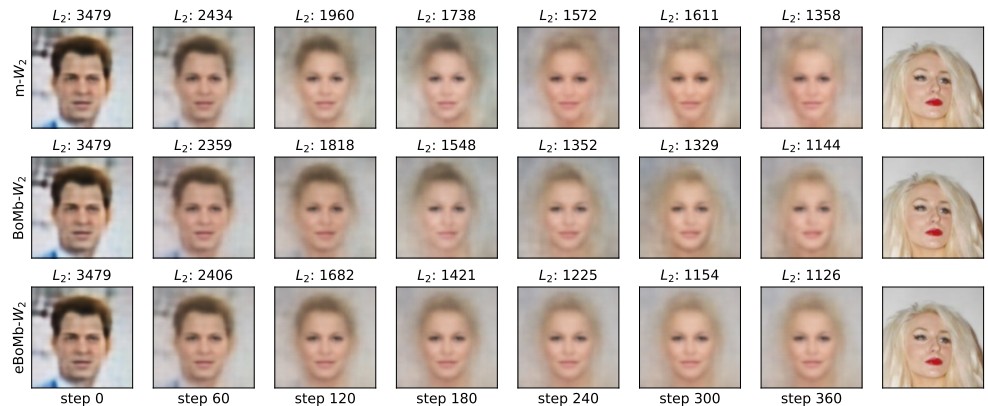

Figure 14: Transforming man face to woman face by using the gradient flow with m-OT and (e)BoMb-OT on CelebA dataset for $(k, m) = (25, 100)$. The last column is the nearest image (in sense of $L_2$ distance) to the final found female image, the corresponding $L_2$ distance are also shown on the top of middle-stage images.

$\tau = 50$ for CIFAR10, $\tau = 40$ for CelebA. For the eBoMb-OT, we choose the best setting for $\lambda \in \{1, 2, 3, 4, 5, 10, 20, 30, 40, 50, 60, 70, 80\}$

**Neural network architectures:** We use the MLP for the generative model on the MNIST dataset, while CNNs are used on the CelebA dataset.

Generator architecture was used for MNIST dataset:
$z \in \mathbb{R}^{32} \to FC_{100} \to ReLU \to FC_{200} \to ReLU \to FC_{400} \to ReLU \to FC_{784} \to ReLU$

Generator architecture was used for CelebA: $z \in \mathbb{R}^{128} \to TransposeConv_{512} \to BatchNorm \to ReLU \to TransposeConv_{256} \to BatchNorm \to ReLU \to TransposeConv_{128} \to BatchNorm \to ReLU \to TransposeConv_{64} \to BatchNorm \to$

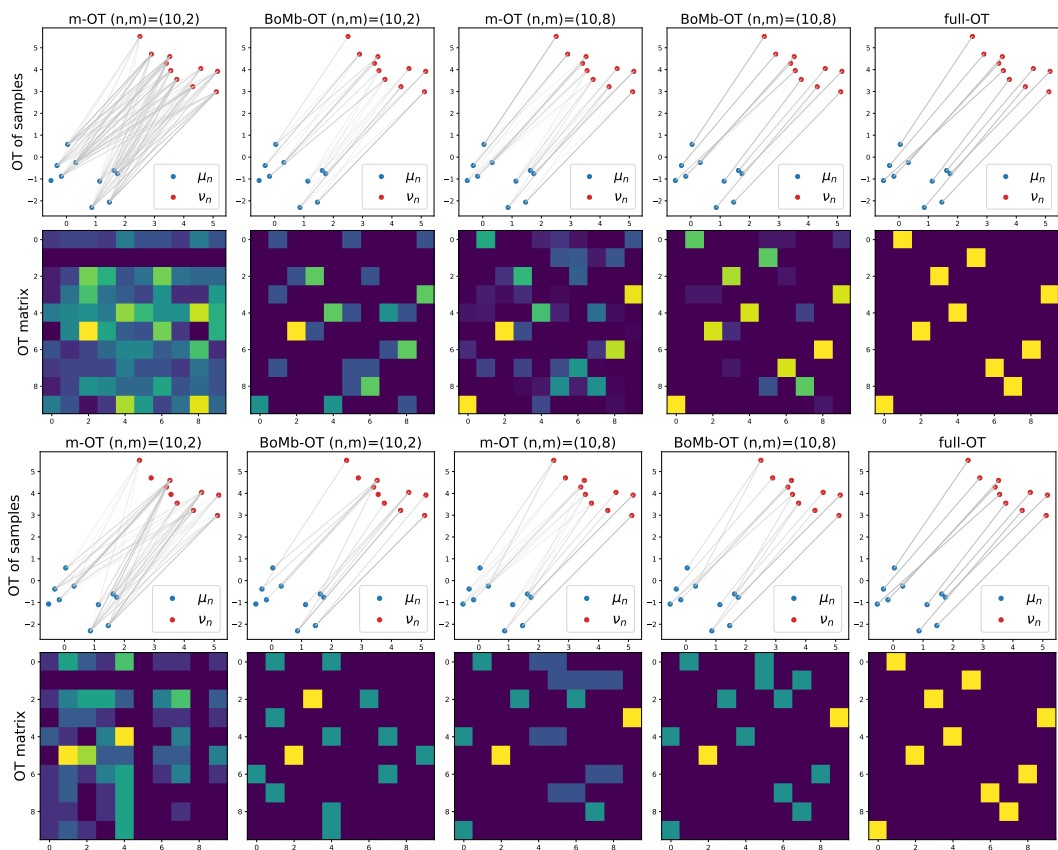

Figure 15: Visualization of transportation plans of m-OT and BoMb-OT between 2D empirical distributions with 10 samples. The first two rows are results of $k = 20$. The next two rows are for $(k, m) = (8, 2)$ and $(k, m) = (2, 8)$ in turn.

$ReLU \rightarrow TransposeConv_1 \rightarrow Tanh$
Metric learning neural network's architecture was used for CelebA:

First part: $x \in \mathbb{R}^{64 \times 64 \times 3} \rightarrow Conv_{64} \rightarrow LeakyReLU_{0.2} \rightarrow Conv_{128} \rightarrow BatchNorm \rightarrow LearkyReLU_{0.2} \rightarrow Conv_{256} \rightarrow BatchNorm \rightarrow LearkyReLU_{0.2} \rightarrow Conv_{512} \rightarrow BatchNorm \rightarrow Tanh$
Second part: $Conv_1 \rightarrow Sigmoid$
Generator architecture was used for CIFAR10: $z \in \mathbb{R}^{128} \rightarrow TransposeConv_{256} \rightarrow BatchNorm \rightarrow ReLU \rightarrow TransposeConv_{128} \rightarrow BatchNorm \rightarrow ReLU \rightarrow TransposeConv_{64} \rightarrow BatchNorm \rightarrow ReLU \rightarrow TransposeConv_1 \rightarrow Tanh$
Metric learning neural network's architecture was used for CIFAR:

First part: $x \in \mathbb{R}^{32 \times 32 \times 3} \rightarrow Conv_{64} \rightarrow LeakyReLU_{0.2} \rightarrow Conv_{128} \rightarrow BatchNorm \rightarrow LearkyReLU_{0.2} \rightarrow Conv_{256} \rightarrow BatchNorm \rightarrow Tanh$
Second part: $Conv_1 \rightarrow Sigmoid$

## E.2 DEEP DOMAIN ADAPTATION

In this section we state the neural network architectures and hyper-parameters for deep domain adaptation.

**Evaluation metric:** The classification accuracy is utilized to evaluate the mini-batch methods.

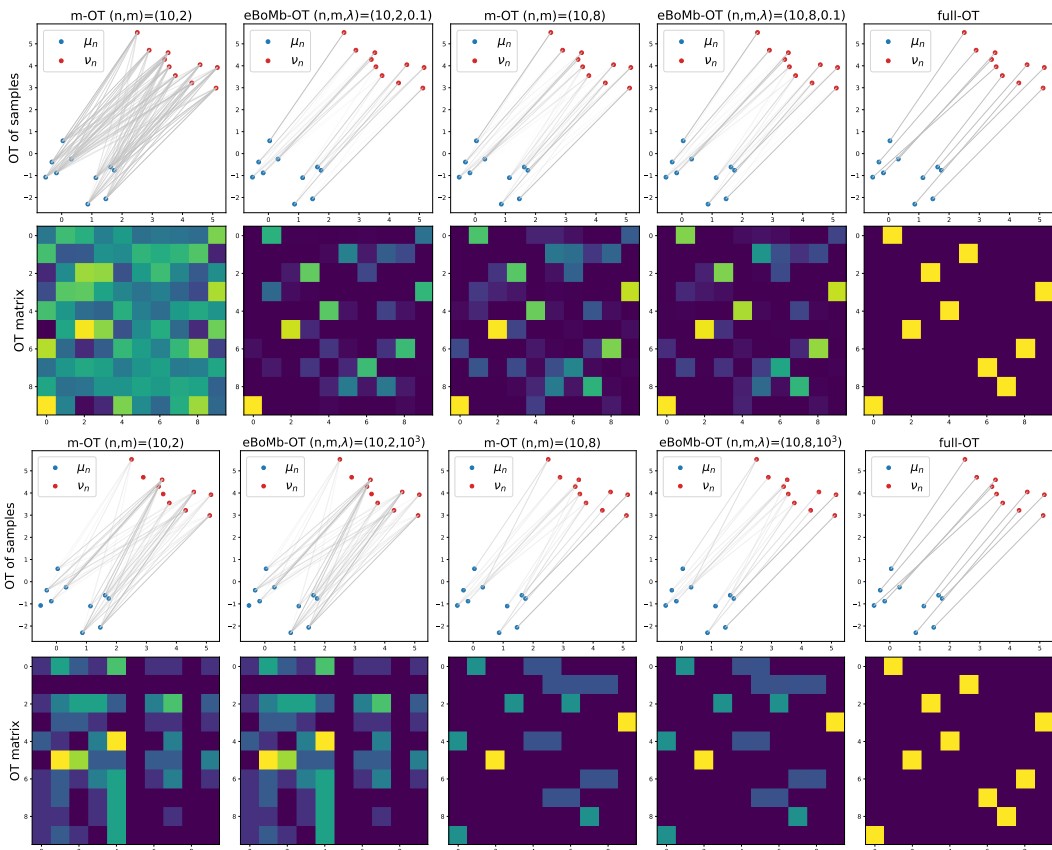

Figure 16: Visualization of the m-OT's transportation plan and eBoMb-OT's transportation plan between two 2D empirical distributions with 10 samples, and the interpolation property of the eBoMb-OT. The first four rows provide the result for eBoMb-OT ($\lambda \in \{0.1, 10^3\}$), and $k = 40$. The last row is the transportation plan with $(k, m) = (8, 2)$ and $(k, m) = (2, 8)$ respectively.

**Parameter settings for digits datasets:** The number of mini-batches $k$ varies in $\{8, 16, 32\}$. For the SVHN dataset, the mini-batch size $m$ is set to $50$. Because the USPS dataset has fewer samples than the SVHN dataset, $m$ is set to $25$. We train the network using Adam optimizer with an initial learning rate of $0.0002$. The number of epochs is $80$ for $k = 8$. As the number of mini-batches doubles, we double the number of epochs so that the number of iterations does not change. The hyperparameters of m-OT and BoMb-OT in Equation 10 follow the settings in DeepJDOT: $\alpha = 0.001, \lambda_t = 0.0001$. The hyperparameters of m-UOT and eBoMb-UOT are the same as JUMBOT: $\alpha = 0.1, \lambda_t = 0.1, \epsilon = 0.1, \tau = 1$. For computing eBoMb-UOT, we choose the best value of $\lambda \in \{0.01, 0.1, 0.2, 1, 10\}$ (entropic regularizer coefficient).

**Parameter settings for Office-Home dataset:** The number of mini-batches $k$ varies in $\{2, 4, 8\}$. Following the settings in (Fatras et al., 2021a), we train models with a mini-batch size $m = 65$ during $10000$ iterations. The hyperparameters for computing the cost matrix follow the settings in JUMBOT: $\alpha = 0.01, \lambda_t = 0.5, \tau = 0.5, \epsilon = 0.01$. For computing eBoMb-UOT, we choose the best value of $\lambda \in \{0, 0.01, 1, 100\}$ (entropic regularizer coefficient).

**Training details:** Similar to both DeepJDOT and JUMBOT, we stratify the data loaders so that each class has the same number of samples in the mini-batches. For digits datasets, we also train our neural network on the source domain during $10$ epochs before applying our method. For Office-home dataset, because the classifiers are trained from scratch, their learning rates are set to be $10$ times that of the generator. We optimize the models using an SGD optimizer with momentum $= 0.9$ and weight decay $= 0.0005$. We schedule the learning rate with the same strategy used in JUMBOT. The learning

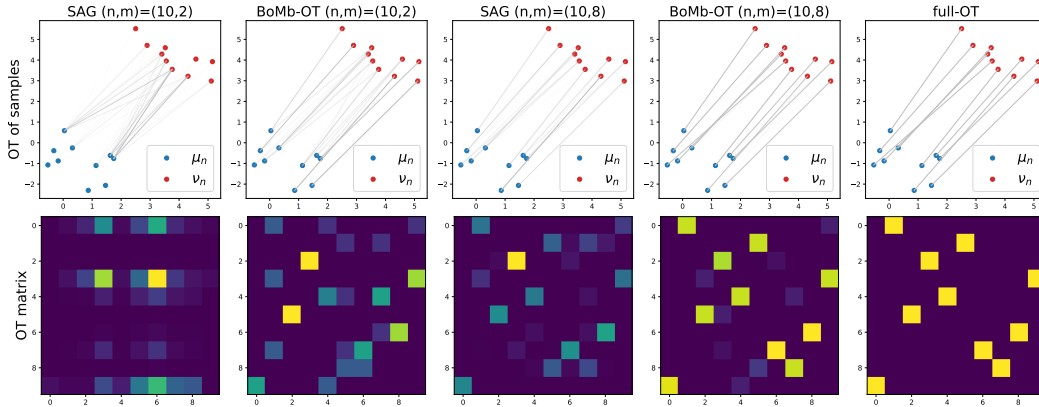

Figure 17: Visualization of the transportation plan of the BoMb-OT and the stochastic averaged gradient (SAG) method between two 2D empirical distributions with 10 samples. The number of mini-batch for the BoMb-OT is set to 30. the learning rate for SAG is the best in $\{0.001, 0.005, 0.01, 0.05, 0.1, 0.5, 1, 5, 10\}$, the entropic regularization of SAG is the best in $\{0.01, 0.05, 0.1, 0.5, 1, 5, 10\}$.

rate at iteration $p$ is $\eta_p = \frac{\eta_0}{(1+\mu q)^\nu}$, where $q$ is the training progress linearly changing from 0 to 1, $\eta_0 = 0.01, \mu = 10, \nu = 0.75$.

**Neural network architectures:** On digits datasets, we use CNN for our generator and 1 FC layer for our classifier in both adaptation scenarios. For Office-Home dataset, our generator is a ResNet50 pre-trained on ImageNet except for the last FC layer, which is our classifier.

Generator architecture was used for SVHN dataset:
$z \in \mathbb{R}^{32 \times 32 \times 3} \rightarrow Conv_{32} \rightarrow BatchNorm \rightarrow ReLU \rightarrow Conv_{32} \rightarrow BatchNorm \rightarrow ReLU \rightarrow MaxPool2D \rightarrow Conv_{64} \rightarrow BatchNorm \rightarrow ReLU \rightarrow Conv_{64} \rightarrow BatchNorm \rightarrow ReLU \rightarrow MaxPool2D \rightarrow Conv_{128} \rightarrow BatchNorm \rightarrow ReLU \rightarrow Conv_{128} \rightarrow BatchNorm \rightarrow ReLU \rightarrow MaxPool2D \rightarrow Sigmoid \rightarrow FC_{128}$

Generator architecture was used for USPS dataset:
$z \in \mathbb{R}^{28 \times 28 \times 3} \rightarrow Conv_{32} \rightarrow BatchNorm \rightarrow ReLU \rightarrow MaxPool2D \rightarrow Conv_{64} \rightarrow BatchNorm \rightarrow ReLU \rightarrow Conv_{128} \rightarrow BatchNorm \rightarrow ReLU \rightarrow MaxPool2D \rightarrow Sigmoid \rightarrow FC_{128}$

Classifier archiecture was used for both SVHN and USPS datasets:
$z \in \mathbb{R}^{128} \rightarrow FC_{10}$

### E.3 GRADIENT FLOW

For the implementation of the gradient flow, we use the geomloss library (Feydy et al., 2019). The learning rate is set to 0.001. For the autoencoder in CelebA experiments, we use the repo in `https://github.com/rasbt/deeplearning-models/blob/master/pytorch_ipynb/autoencoder/ae-conv-nneighbor-celeba.ipynb` for the pre-trained autoencoder.

### E.4 COMPUTATIONAL INFRASTRUCTURE

All deep learning experiments are done on a RTX 2080 Ti GPU and a GTX 1080 Ti. Other experiments are done on a MacBook Pro 11inc M1.

