# OpenReview forum: "On Transportation of Mini-batches: A Hierarchical Approach"
_ICLR.cc/2022/Conference — ICLR 2022 Submitted_

### Official Review · Reviewer_ScyT · 2021-11-02

**Correctness:** 3
**Technical Novelty And Significance:** 3
**Empirical Novelty And Significance:** 2
**Recommendation:** 3
**Confidence:** 4

**Main Review:**

The paper describes correctly their new loss function as well as some weaknesses from the original minibatch formulation. They provide small toy examples to see the differences between the proposed and the original losses. They also discuss the computational complexity of their method and discuss some practical aspects of their contributions.

Evaluation: My feeling is that the theoretical and experimental aspects are not studied enough.

For instance on the theoretical side:

(i) While discussing the connections between samples from the transport plan, the authors do not discuss the transported mass.

(ii) Also a study similar to [1] about the possible bias of stochastic gradient is important. Would minimizing your proposed loss function with SGD lead to the correct minimum ?


On the experimental side, there are many experiments, but some of them are not extensive (generative modelling and domain adaptation) which make the performances of their method unclear. I am also skeptical that the method is really competitive in practice (numerically and memory) with state of the art method due to the fact that it needs a bigger k for the loss to be computed.

(i) I would encourage the authors to compelete their work on generative modelling and domain adaptation with comparison with recent methods, including methods which are also not based on optimal transport.

(ii) On domain adaptation and generative modelling, I am not convinced that the proposed method outperforms other methods in practice due to its overall complexity (k>1). Other methods rely on a k equals to 1 (including deepjdot) which might make them faster to train and as efficient or more.

(iii) On domain adaptation, there are more recent methods and I would like to see the comparison [2,3]. There is also this paper [4] that you cite in your supplementary and that you should compare to as this paper also tries to minimize the impact of undesirable connections from minibatches.

Related Work and Discussion: The related work is complete and well discussed. The bibliography style seems respected.

Clarity: The paper is clearly written except for the experimental part. A reader which do not know about generative modelling, domain adaptation, color transfer or bayesian computation can not understand the purpose of the different experiments as the explanations are in the supplementary materials. This gives the feeling that the authors just stacked experiments without discussing them properly and entirely. I think this is mostly due to the high number of different experiments. I suggest to focus on two different experiments, to develop the original problem, their methods to solve them and, finally to produce extensive experiments of different methods on several datasets. The remaining experiments could be added to the supplementary.

Questions and remarks:
1. Does your loss transport all samples ? I think it does, but it should be proven. An overall theoretical study of the transport plan is lacking.
2. Is your method an upper bound of OT and a lower bound of minibatch OT ?
3. Time comparison between your method and original minibatch OT. This could be done on domain adaptation experiments for instance.
4. Name of tables are unclear. Please add the meaning of the scores you five (Table 1 and 2).

[1] Learning with minibatch Wasserstein: asymptotic and gradient properties, Fatras et al.
[2] Sliced Wasserstein Discrepancy for Unsupervised Domain Adaptation, Lee et al.
[3] Reliable Weighted Optimal Transport for Unsupervised Domain Adaptation, Xu et al.
[4] Unbalanced minibatch Optimal Transport; applications to Domain Adaptation, Fatras et al.


------------------------------
Rebuttal
------------------------------

Thank you for your detailed answers to my questions. I have read the other reviews and the different answers. I still think that the idea is appealing, but the way the paper is written and the fact that the experiments are not extensive enough prevent a publication at that time in my opinion. I keep my score unchanged and I would encourage the authors to pursue the experimental evaluations or theoretical evaluations (distance of the plan marginals to the marginals ?) of their method.

**Summary Of The Paper:**

Goals: This paper introduces a new optimal transport loss based on a minibatch computation in order to alleviate some weaknesses from the original minibatch OT formulation. The formulation treats minibatches as data and seek to transport the minibatches from the source distribution to the minibatches from the target distribution. By doing so, their method prevents some undesirable connections between data.

**Summary Of The Review:**

The proposed formulation is appealing and I like the idea. However, in my opinion, some theoretical and experimental aspects are lacking, and that prevent a publication at the moment (see the different points above). There is a big discussion regarding where the method can applied and it could be replaced by a discussion on theoretical aspects which are not mentionned. Regarding experiments, I encourage the authors to focus on two experiments (Generative modelling and domaine adaptation) and to make extensive experiments with recent baselines.

---

> ### Author Response · Authors · 2021-11-16
> **Response to Reviewer ScyT: Part 1**
>
> **Q1**: While discussing the connections between samples from the transport plan, the authors do not discuss the transported mass.
>
> **Answer**: Thanks for your question. We assume that your question is about the concentration of the transportation plan of BoMb-OT. We would like to note that theoretical analysis on the concentration of transportation plans of optimal transport is a challenging problem. To the best of our knowledge, there is only a few recent progress on this topic, such as [R.1], [R.2], [R.3]. The common spirits of the results from these papers can be summarized as follows: assume that we have data $X_{1}, \ldots, X_{n}$ that are i.i.d. from distribution $\mu$ and data $Y_{1}, \ldots, Y_{n}$ that are i.i.d. from distribution $\nu$. We respectively denote by $\mu_{n}$ and $\nu_{n}$ the empirical distributions of $X_{1}, \ldots, X_{n}$ and $Y_{1}, \ldots, Y_{n}$. In [R.1], the authors demonstrated that rate of convergence of the transportation map of $W_{2}^2(\mu_{n}, \nu_{n})$ to that of $W_{2}^2(\mu, \nu)$ is similar to the rate of convergence of $W_{2}(\mu_{n}, \nu_{n})$ to $W_{2}(\mu, \nu)$ when the Breiner potential from $\mu$ and $\nu$ is strongly convex (cf. Proposition 12 in [R.1]).
>
> For the proposed BoMb-OT, it can be thought of as a two-layer Wasserstein metric, namely, the first layer is the Wasserstein metric between empirical measures of two mini-batches and the second layer is Wasserstein metric to match mini-batches. Given the hierarchical structures of BoMb-OT, theoretical analysis on the BoMb-OT transportation plan is more challenging than that of OT. It is also the main reason that we have only provided empirical results for the BoMb-OT transportation plan in the current paper. Here, we also would like to highlight a possible approach to theoretically understand the convergence of the BoMb-OT transportation plan. Recall that, we denote $\hat{\pi}^m_k(\mu_n,\nu_n)$ the BoMb-OT transportation plan in Definition 3.
>
> Our possible approach consists of two steps. For the first step, we need to prove that $\hat{\pi}^m_k(\mu_n,\nu_n)$ converges to $\hat{\pi}(\mu, \nu)=E_{\left(X^{m}, Y^{m}\right) \sim \hat{\gamma}}\left(\pi_{X^{m}, Y^{m}}\right)$ where $\hat{\gamma}$ is the optimal transportation plan of population BoMb-OT  $D_d^m(\mu,\nu)$ and $\pi_{X^{m}, Y^{m}}$ is the optimal transportation plan of computing $d(P_{X^m},P_{Y^m})$. If we further would like to understand how close BoMb-OT's transportation plan is to the transportation plan of OT, we need to understand the gap between $\hat{\pi}(\mu, \nu)$ and the optimal transportation plan between $\mu$  and $\nu$. Given the technical complication of each step, we defer detailed developments of these steps in the future work.
>
> References:
>
> [R.1] T. Manole, S. Balakrishnan, J. Weed, L. Wasserman. Plugin Estimation of Smooth Optimal Transport Maps. Arxiv Preprint Arxiv: 2107.12364, 2021
>
> [R.2] J. C. Hutter, P. Rigollet. Minimax rates of estimation for smooth optimal transport maps. Annals of Statistics, 2021
>
> [R.3] N. Deb, P. Ghosal, and B. Sen. Rates of estimation of optimal transport maps using plug-in estimators via barycentric projections. Arxiv Preprint Arxiv: 2107.01718, 2021
>
> **Q2**: Also a study similar to [1] about the possible bias of stochastic gradient is important. Would minimizing your proposed loss function with SGD lead to the correct minimum?
>
> **Answer**: We would like to remark that the unbiased stochastic gradient m-OT leads to the local minimum of the population m-OT, which is not a metric. Therefore, it is not clear whether that minimum can say anything about the closeness of the two measures. On the other hand, as we have shown in the paper, population BoMb-OT is a proper metric between two probability measures, which can guarantee that when that metric is small the two probability measures will be close. In our experiments with empirical BoMb-OT, we have shown that for deep learning applications such as deep generative models, deep domain adaptation, using stochastic gradients of BoMb-OT can improve the performance of the trained neural nets. It may imply that our proposed loss function with SGD converges to a more favorable local minimum than that of m-OT; however, a rigorous theory for that point is challenging and we leave this question for future investigation.

---

> > ### Comment · Reviewer_ScyT · 2021-11-26
> > **Thank you for your detailed answers**
> >
> > Dear authors,
> >
> > 1. I think there is a misunderstanding and I apologize for not having made my question clearer. I was not talking about the concentration of the transportation plan but either the distance between your minibatch transport plan and the input marginals. I think that when you consider all possible minibatches, you would get the input marginals. In my opinion, it is an important result, as it would show that you approximate a transport problem.
> >
> > 2. Yes I agree that we do not know the quality of the reached minimum with m-ot. And as you said, your method is a metric (remark: I have not been able to check the proof), so I was wondering if your method could have the best of both worlds (metric property and unbiased gradients). I think it is an important question but I understand that it is a challenging.

---

> > > ### Author Response · Authors · 2021-11-28
> > > **Response to Reviewer ScyT**
> > >
> > > 1. We thank you for your responses. Our response also sheds some light on the input marginals. Since we have an additional layer for the optimal transport between mini-batches, obtaining an upper bound for the BoMb's minibatch transport plan and the input marginals requires several new proof techniques as we need to take into account the structures of the optimal transport between mini-batches, which is non-trivial. Note that, the standard concentration tools used in establishing the concentration of m-OT's transportation plan around the input marginals are no longer sufficient for our case. Given the non-triviality in proof techniques, we would like to leave it for the future direction.
> > >
> > > 2. We thank you for your comment. As we have shown in the paper, the population of BoMb-OT is a proper metric in the probability space (cf. Theorem 1). For the gradients,  the stochastic gradient that is estimated by BoMb-OT loss is biased to the population BoMb-OT.  However, we can derive an extension of the population BoMb-OT that we can derive an unbiased gradient estimator by using the debiasing method in [5].
> > >
> > > [5] Tim Salimans, Han Zhang, Alec Radford, Dimitris Metaxas, "Improving GANs Using Optimal Transport, Tim Salimans", ICLR 2018.

---

> ### Author Response · Authors · 2021-11-16
> **Response to Reviewer ScyT: Part 2**
>
> **Q3**: I would encourage the authors to complete their work on generative modelling and domain adaptation with comparison with recent methods, including methods which are also not based on optimal transport.
>
> **Answer**: Thank you for your suggestion. Our experiments are designed to show that BoMb-OT works better than m-OT in three types of applications: gradient-based, mapping-based, and discrepancy-based applications. Obtaining state-of-the-art performance for deep learning applications requires a lot of engineering work including neural network architecture selection, implementation tricks, and additional losses. We believe that it is beyond the scope of our current paper. Also, since optimal transport can be used for many other large-scale non-deep learning applications, we believe that it is relevant to study the performance of BoMb-OT on these applications.
> Moreover, our current experiments on generative modeling have already shown the favorable performance of BoMb-OT. In particular, we have run generative models that use sliced Wasserstein distance and the Sinkhorn divergence on standard datasets including MNIST, CIFAR10, and CelebA. We would like to remark again that our paper does not aim to propose a new domain adaptation method or a new generative modeling method. Indeed, we focus on proposing a new minibatch framework, BoMb-OT, which is potentially useful for many frameworks of domain adaptation.
>
> **Q4**: On domain adaptation and generative modelling, I am not convinced that the proposed method outperforms other methods in practice due to its overall complexity (k>1). Other methods rely on a k equals to 1 (including deepjdot) which might make them faster to train and as efficient or more.
>
> **Answer**: Thanks for your question. Indeed, there is a trade-off here. In particular, BoMb-OT is the option when people want to have high performance in applications. Furthermore, BoMb-OT can be implemented in parallel so it can exploit the benefit of multi-devices training. On the other hand, despite its fast and efficient training, using optimal transport loss with k=1 suffers from non-optimal matchings between samples. BoMb-OT gives a new option for practitioners to deal with that issue when they want to design minibatch OT losses.
>
> **Q5**: On domain adaptation, there are more recent methods and I would like to see the comparison [2,3]. There is also this paper [4] that you cite in your supplementary and that you should compare to as this paper also tries to minimize the impact of undesirable connections from minibatches.
>
> **Answer**: Thank you for the references, we have cited these papers in our revised version. For [2], [3], their frameworks for domain adaptation are different from the framework that we followed from DeepJDOT. The BoMb-OT scheme can also be applied to those frameworks; however, it requires a lot of extra work to reproduce the codes of those frameworks which are unavailable. Therefore, we believe it is out of the scope of the paper.
>
> For [4], the authors change the type of transportation metric between mini-batch measures from OT to UOT. As discussed in the paper, a mini-batch scheme can be decomposed into two steps. The first step solves transportation between mini-batch measures and the second step aggregates them into global transportation. BoMb-OT proposes to use an additional OT in the second step to combine mini-batch transportation, so we can directly extend BoMb-OT to BoMb-UOT and eBoMb-UOT. Moreover, we want to note that we can also utilize UOT in the second step of the proposed mini-batch scheme; however, we believe that this is an orthogonal direction. We are running experiments to compare (e)BoMb-UOT with m-UOT on digit datasets and office home datasets. We will include the experimental results when we finish the experiments.

---

> > ### Author Response · Authors · 2021-11-22
> > **Additional Experiments for Question 5**
> >
> > Thank you for your comments on Question 5. We have finished some additional experiments on domain adaptation for comparing m-UOT and eBoMb-UOT.  We used the number of mini-batches $k \in \\{ 8, 16, 32\\}$ on digit datasets and $k \in \\{2, 4, 8\\}$ on the Office-home dataset. For parameters of UOT, we set the coefficient of the entropic regularizer $\epsilon$ to 0.1 on digit-datasets and to $0.01$ on Office-home datasets. We set the marginal regularizer coefficient $\tau$ to $1$ on digit datasets, and to $0.5$ on Office-home datasets.  We would like to recall that we use the best parameter setting of UOT that was reported in [4]. We choose the best entropic regularizer coefficient $\lambda$ for eBoMb-UOT in the set $\\{0, 0.01, 1, 100\\}$. For more details of other hyperparameters and the training process, we refer to the revised Appendix E.2 which is colored in blue. We have updated the classification accuracy on digit datasets in Table 2  in the main text of the paper and classification accuracy on the Office-home dataset in Table 5 in Appendix D.2.  From these tables, eBoMb-UOT improves slightly the classification accuracy on the target domain on both types of datasets.  For example, eBoMb-UOT has 0.24 higher classification accuracy on average on Office-home datasets and 0.32 higher classification accuracy on USPS-MNIST adaptation. We will continue to conduct the experiments during the discussion time.

---

> > ### Comment · Reviewer_ScyT · 2021-11-26
> > **Answer to Q3, Q4 and Q5**
> >
> > 3. On this point, I disagree with the authors. The authors said in their paper: "First, the BoMb-OT could provide a more similar transportation plan to the original OT than the m-OT, which leads to a more meaningful discrepancy using mini-batches". Ok, but your method is not the only method which tries to have a more meaningful minibatch transport plan. It thus needs to be compared with other minibatch OT variants which look for a better transportation plan [3,4 on domain adaptation for instance and that you have done with in your revised paper]. I understand that the theory is challenging, but then you should demonstrate your different claims experimentally and compare with methods which have the same motivations as yours.
> >
> > 4. Thank you for your answer but I am not convinced. "BoMb-OT can be implemented in parallel so it can exploit the benefit of multi-devices training" so does m-ot or m-uot. You also say "using optimal transport loss with k=1 suffers from non-optimal matchings between samples.", yes but several variants were published to mitigate this problem and you should compare your proposed method with them. This is a problem in my opinion, because we can not know if your proposed method is better than them and according to your knew experiments, your proposed method, when it uses exact OT, does not compete with [4], so I clearly wonder the quality of the estimated transport plan.
> >
> > 5. Thank you for the additionnal experiment with [4]. As I said in 4., I am a bit skeptical with the proposed method and its associated transport plan. It might be indeed be more insightful than the one from m-ot but clearly does not compete with m-uot, while m-uot is not a metric. I agree with you that you can adapt your method to UOT and it is interesting to see that you get similar results as [4] without hypertunning the different hyper-parameters. It is a promising research direction in my opinion.

---

> > > ### Author Response · Authors · 2021-11-27
> > > **Response to Reviewer ScyT**
> > >
> > > Thank you for your comments. We agree that m-UOT [4] can avoid cross-cluster connections that improve significantly the result of domain adaptation, and the transportation plan of BoMb-OT is not as good as m-UOT. However, as in our paper's title, we propose a hierarchical approach for mini-batch optimal transport that can be adapted for all types of transportation $d$ between mini-batch measures ,e.g., OT, UOT, and their variants ([3]). The comparison that we did is between weighted averaging with an additional OT and conventional averaging, it is not between any specific choice of transportation types $d$.
> > >
> > > The reason we focus on OT is that OT has been utilized successfully in many applications e.g. deep generative model, approximate Bayesian computation while there is still a challenge for applying UOT. For example, the training generative model with UOT needs to deal with the significant change of the scale of sample cost matrices that require adaptively choosing the marginal relaxation parameter $\tau$. In approximate Bayesian computation, the discrepancy for accept-reject criteria prefers more metricity for the correct decision of taking samples and throwing them away. From this aspect, UOT is not appealing as OT. As a reference, authors in [3] did not compare m-UOT with m-OT on deep generative models, approximate Bayesian computation, and color transfer.
> > >
> > > In summary, we would like to remark that our contribution is independent of the choice of transportation between mini-measures. We have shown that the hierarchical approach of BoMb is better than the conventional averaging scheme in various applications. We focus mostly on OT since the OT has been widely used in several applications. Furthermore, as we already included in the revision, our initial experiments for using BoMb with UOT (without any attempt to search for the optimal hyperparameters of UOT) in deep domain adaptation showed some slight improvement over m-UOT. Therefore, we believe that the hierarchical approach helps improve the minibatch optimal transport frameworks in general.
> > >
> > > We are happy to discuss if you still have other concerns.

---

> ### Author Response · Authors · 2021-11-16
> **Response to Reviewer ScyT: Part 3**
>
> **Q6**: The paper is clearly written except for the experimental part. A reader which does not know about generative modelling, domain adaptation, color transfer or bayesian computation can not understand the purpose of the different experiments as the explanations are in the supplementary materials. This gives the feeling that the authors just stacked experiments without discussing them properly and entirely. I think this is mostly due to the high number of different experiments. I suggest focusing on two different experiments, to develop the original problem, their methods to solve them and, finally, to produce extensive experiments of different methods on several datasets. The remaining experiments could be added to the supplementary.
>
> **Answer**: Thank you for your suggestion. We believe that discussing multiple types of applications could push forward the usage of optimal transport in the community. For that reason, in Section 3.1, we discuss gradient-based applications, mapping-based applications, and discrepancy-based applications. Finally, we would like to repeat that our paper is to propose a new mini-batch OT scheme that can be useful for several machine learning and statistics applications. We do not aim to particularly focus on any particular applications and we want to show that the new scheme can be applied to various situations and is better than the conventional scheme.
>
> **Q7**: Does your loss transport all samples? I think it does, but it should be proven. An overall theoretical study of the transport plan is lacking.
>
> **Answer**: Please refer to our answer to Q.1 for the theoretical study of the transport plan. Furthermore, mini-batch OT losses are used in the process of interactively training a model. So, only a small number of samples are transported at each iteration; however, in the entire process, all samples are transported. For example, some mini-batches are used to estimate the gradient of neural networks in each stochastic gradient descent update in deep learning. Similarly, transportation plans from mini-batches are used to iteratively update a small part of the color palette of the source image. Also, the criteria to accept posterior samples from approximate Bayesian computation is computed via some mini-batches in each iteration.
>
> **Q8**: Is your method an upper bound of OT and a lower bound of minibatch OT?
>
> **Answer**:  We assume that you refer to the population versions of m-OT and BoMb-OT. From the Definition 4 of the population BoMb-OT, it is clear that the BoMb-OT is the lower bound of m-OT as we can just choose the transportation plan $\gamma$ in that definition to be a product measure between $\mu^{\otimes m}$ and $\nu^{\otimes m}$ . For the OT, we assume that the underlying metric $d = W_{p}$ where $p \geq 1$. Then, using the argument in the proof of part (i) of Theorem 2, we have $d(P_{X^{m}}, P_{Y^{m}}) \leq \frac{1}{m^{1/p}} \sum_{i = 1}^{m} \|X_{i} - Y_{i}\|$. This inequality demonstrates that BoMb-OT between $\mu$ and $\nu$ is upper bounded by $m^{1-1/p} W_{1}(\mu, \nu)$.

---

> > ### Comment · Reviewer_ScyT · 2021-11-26
> > **Q6-7-8**
> >
> > Q6. Thank you for your answer. I understand your motivation. However, my feeling is still that you could report these (interesting) sections to appendix and argument that your method brings better transport plan than m-ot and its variants in the main paper.
> >
> > Q7 See my answer for Q1.
> >
> > Q8 Okay, thank you for your answer. I think this is interesting and it should be added to the manuscript.

---

> > > ### Author Response · Authors · 2021-11-27
> > > **Response to Reviewer ScyT**
> > >
> > > Thank you for your comments. We will add these discussions to the revised version of our paper.

---

> ### Author Response · Authors · 2021-11-16
> **Response to Reviewer ScyT: Part 4**
>
> **Q9**: Time comparison between your method and original minibatch OT. This could be done on domain adaptation experiments for instance.
>
> **Answer**: We have already provided the computational speed for deep generative models in Table 3 in Appendix D.1, the computational speed for deep DA in Tables 4-6 in Appendix D.2, and the computational speed for color transfer in Appendix  D.3. We observe that BoMb-OT runs faster than m-OT in deep learning applications. In particular, the most expensive task in deep DA and deep generative models is to estimate the gradient of the neural network, namely, backpropagating through the corresponding computational graphs. Since m-OT treats every pair of mini-batches the same, it needs to cumulate the gradient from all mini-batches. On the other hand, the additional OT of BoMb-OT can discard some pairs of mini-batches that are extremely different by assigning 0 weight to those pairs. As a result, we do not need to do gradient estimation for those mini-batches in BoMb-OT, hence, the BoMb-OT could be faster than m-OT in deep learning applications. In more detail, for m-OT, we have to estimate the gradient from all $k^2$ pairs of mini-batches while BoMb-OT only needs to estimate for $k$ pair of mini-batches. The reason is that there are exactly $k$ entities $\gamma_{ij}$ that are non-zeros in the optimal transportation plan between mini-batches. The number of non-zero entries increases when we use eBoMb-OT however it is still smaller than the number of m-OT. For more information, we refer the reviewer to look at our pseudo-codes for each application in the appendix, namely, Algorithm 1 in Appendix C.1. for deep generative models, Algorithm 2 in Appendix C.2. for deep domain adaptation, Algorithm 3 in Appendix C.3 for color transfer, and Algorithm 4 in Appendix C.4 for approximate Bayesian computation.
>
> **Q10**: Name of tables are unclear. Please add the meaning of the scores you five (Table 1 and 2).
>
> **Answer**: Thank you for your comment. As mentioned in the experimental part, in Table 1, we reported the approximated $W_2$ distance for MNIST and FID scores for CIFAR10 and CelebA. In Table 2, we reported the classification accuracy on the target domain. We have added more information to the caption of our tables in the revised version.

---

> > ### Comment · Reviewer_ScyT · 2021-11-26
> > **Q9-Q10**
> >
> > Q9. This is clearly an unexpected result, and I am really positively surprised ! This should be discussed (or at least mentioned clearly) in the main paper. Having a better computational complexity than m-OT, where the motivation of m-ot was to have a better computational (and memory) complexity than OT,  is a very good motivation of using your method.
> >
> > Q10. Thank you. It is clearer now.

---

### Official Review · Reviewer_C31a · 2021-11-03

**Correctness:** 3
**Technical Novelty And Significance:** 3
**Empirical Novelty And Significance:** 3
**Recommendation:** 6
**Confidence:** 3

**Main Review:**

strengths:

The optimal transport (OT) problem studied in this paper is important. Existing work and their advantages and disadvantages are discussed and the study is well motivated.

Theorem 2 shows that the proposed BoMb-OT method is sound and approximating some well-defined metrics. On the other hand, the authors provided the implementation and demonstrated the usefulness of the proposed method using several different applications.

weakness:

1. As mentioned, the population m-OT has some problem of not preserving metric properties. As also noted the entropic regularized version of population BoMb-OT would recover m-OT in some cases. Does this mean the BoMb-OT will also recover the same problem that m-OT would have?

2. Figure 1 is confusing to me. I get the authors wanted to provide some intuitive explanations for the advantages of using BoMb-OT. However, I do not understand why and how using BoMb-OT can achieve the right transportation in the figure while using m-OT cannot. I suggest if possible these examples can come with specific numbers such that we can calculate and verify the results, making them easier to understand and more convincing.

3. The BoMb-OT has one additional OT comparing to m-OT as noted in "Computational complexity of the BoMb-OT" paragraph ($O(k^2(m^2+1)/\epsilon^2)$ vs. $O(k^2 m^2 /\epsilon^2)$). It looks contradiction to what is mentioned in the appendix, i.e., "The run time of the m-OT nearly doubles that of the BoMb-OT". Is this from a better implementation of BoMb-OT. Please clarify this.

**Summary Of The Paper:**

This paper proposed Batch of Mini-batches Optimal Transport (BoMb-OT) method, which finds the optimal coupling between mini-batches in mini-batch optimal transport (m-OT), which is achieved by solving another OT problem over the mini-batches.

The authors claimed that doing this will capture the relation between different mini-batches better. They firstly proved that BoMb-OT approximates the population BoMb-OT metric in probability measure space with and without entropic regularization.

The authors then implemented the proposed BoMb-OT method and applied it on deep generative models and deep domain adaptation, showing that BoMb-OT has favourable performance over m-OT.




**Summary Of The Review:**

Overall, I found the problems studied in this paper interesting and important, the proposed method reasonable, theoretically sound, and well supported by experiments and implementation details.

---

> ### Author Response · Authors · 2021-11-16
> **Response to Reviewer C31a**
>
> **Q1**: As mentioned, the population m-OT has some problem of not preserving metric properties. As also noted the entropic regularized version of population BoMb-OT would recover m-OT in some cases. Does this mean the BoMb-OT will also recover the same problem that m-OT would have?
>
> **Answer**: The entropic population BoMb-OT admits m-OT as its special case when the entropic regularizer coefficient $\lambda$ goes to infinity. This generalization guarantees that the performance of the BoMb-OT is at least comparable to m-OT when the regularizer is sufficiently large. In practice, $\lambda$ is not chosen very big, e.g., 1, 2, 3, ..,10. Using the entropic regularizer benefits from a faster solver with the Sinkhorn algorithm that can reduce the computational complexity from $\mathcal{O}(k^3)$ to $\mathcal{O}(k^2)$ for approximating the optimal transport. Also, an entropic regularizer makes the objective smoother which might be appealing for training some applications such as deep generative models.
>
> **Q2**: Figure 1 is confusing to me. I get the authors wanted to provide some intuitive explanations for the advantages of using BoMb-OT. However, I do not understand why and how using BoMb-OT can achieve the right transportation in the figure while using m-OT cannot. I suggest if possible these examples can come with specific numbers such that we can calculate and verify the results, making them easier to understand and more convincing.
>
> **Answer**: Thank you for the suggestion. In Figure 1, we consider two uniform measures of 3 points. In particular, the first measure has the supports that are red boxes in the two-dimensional spaces with locations (0,3), (1,2), (0,1) respectively, and the second measure has the supports that are blue circles with locations (3,3), (4,2), (3,1) respectively. Solving OT between these two measures yields the transportation plan that is one to one as shown in the figure. With the assumption that we can only solve the OT of size 2 due to the limitation of memory and time, we divide the supports of each measure into two mini-batches of size 2, namely, $A_1$ and $A_2$, $B_1$ and $B_2$. For each mini-batch, we define a uniform measure on its supports (0.5 mass for each point). Since m-OT takes the average of the transportation plan from four pairs of mini-batches ($A_1 - B_1$, $A_1 - B_2$, $A_2 - B_1$, $A_2 - B_2$)  as the global transportation plan, it creates many redundant matchings. In contrast, BoMb-OT considers an additional transportation problem between mini-batches, namely, we define two uniform measures which have supports are mini-batches. The ground metric in this case is the optimal transport metric between corresponding mini-batch measures that we have computed. After this step, we can discard redundant pairs of mini-batches ($A_1 - B_2$ and $A_2-B_1$), which leads to a better global transportation plan.
>
> **Q3**: The BoMb-OT has one additional OT compared to m-OT as noted in "Computational complexity of the BoMb-OT" paragraph ( vs. ). It looks contradiction to what is mentioned in the appendix, i.e., "The run time of the m-OT nearly doubles that of the BoMb-OT". Is this from a better implementation of BoMb-OT. Please clarify this.
>
> **Answer**: Thank you for the interesting question. In deep learning applications, the most expensive task is to estimate the gradient of the neural network, namely, backpropagating through the computational graph. Since m-OT treats every pair of mini-batches the same, it needs to cumulate the gradient from all pairs of mini-batches. On the other hand, the additional OT of BoMb-OT can discard some pairs of mini-batches that are extremely different by assigning 0 weight to those pairs. As a result, we do not need to do gradient estimation for those mini-batches in BoMb-OT. Hence, the BoMb-OT could be faster than m-OT in deep learning applications. In more detail, we would like to recall our algorithms for deep DA with BoMb-OT in Algorithm 2 in Appendix C.2. The bottleneck of both m-OT and BoMb-OT is the process of creating a computational graph and backward propagation on GPU. For m-OT, we have to loop over $k^2$ pairs of mini-batches. Although sharing the same number of loops, BoMb-OT actually runs the bottleneck process just $k$ times because there are exactly $k$ entities $\gamma_{ij}$ that are non-zeros in the optimal transportation plan between mini-batches. When eBoMb-OT is used the number of non-zeros entries will increase; however, that number is still smaller than the number of m-OT.

---

> ### Author Response · Authors · 2021-11-28
> **Response to Reviewer C31a  - Any further questions on our current draft**
>
> We would like to thank you again for your reviews and feedback. We have updated our manuscript and added replies to your comments and questions.
>
> We would appreciate it if you could let us know if our responses have addressed your concerns and whether you still have any other questions on the current draft. We would be happy to do any follow-up discussion or address any additional comments.

---

### Official Review · Reviewer_9B4c · 2021-11-04

**Correctness:** 4
**Technical Novelty And Significance:** 2
**Empirical Novelty And Significance:** 2
**Recommendation:** 5
**Confidence:** 3

**Main Review:**

This paper introduces BombOT, a hierarchical approach to combining mini-batches in order to enhance the approximation quality of mini-batch-based approaches to OT.

I believe it is a nice contribution and the main strength is the comprehensive list of experiments performed, illustrating benefits on several tasks in statistics and machine learning.

I have a few concerns, though

1)The theoretical findings are not very strong. Showing that there is a distance being defined is a nice finding, and contrasting this with the lack of a metric structure in usual m-OT suggest that this construction is in the right direction. However, practitioners are more interested in convergence properties. In this regard, theorem 2 doesn't say much, as it corresponds to a comparison between batch and population versions of the proposed method. This doesn't say much about how does this relate to the original wasserstein/sliced/etc distance.

2)Similarly, experiments mainly comparse m-OT and bomb-OT. Comparisons with true Wasserstein distance are mostly lacking. We should be able to at least empirically how large batches need to be so that we will get decent approximations. As a practitioner who tried used m-OT I believe the problem is super important, but the numbers provided don't say much if they are not put into perspective with respect to the true distance. The findings reporting in Fig 3 are honest, but a bit discouraging in that respect. What is the dimension here? how large need k and m need to be so we get a reasonable approximation? (related to (1)).

3)another suggestion (optional): I believe this method can be useful for aligning spaces, as in https://arxiv.org/abs/1809.00013. It would be good to have comparisons there as well, and to show whether this mini-batch approximation can be reasonable (i.e. compare against the no-mini batch case)

**Summary Of The Paper:**

An enhancement over a mini-batch version of OT that provides better empirical evaluation over a large array of tasks

**Summary Of The Review:**

Nice article. But requires a more thorough comparison with true wasserstein.

---

> ### Author Response · Authors · 2021-11-16
> **Response to Reviewer 9B4c: Part 1**
>
> **Q1**: “Practitioners are more interested in convergence properties… This doesn't say much about how does this relate to the original wasserstein/sliced/etc distance.”
>
> **Answer**: Thanks for your insightful comment. We would like to clarify that our purpose of proposing BoMb-OT is not to approximate the full-batch OT. As we mentioned in the paper, the full-batch OT is computationally expensive, namely, its computational complexity is of the order of $\mathcal{O}(n^2/\epsilon)$ and the memory complexity that is at least $\mathcal{O}(n^2)$ with $n$ is the number of supports. Therefore, the full-batch OT cannot be used in large-scale machine learning settings due to the memory issue.
>
> In the paper, we would like to propose BoMb-OT as an alternative for full-batch OT in large-scale machine learning applications. Previously, people used m-OT as a standard mini-batch optimal transport framework in these large-scale applications; however, as we pointed out in the paper, the population loss of m-OT does not define a proper metric and is not zero when the two probability measures are equal. Furthermore, it is also unclear whether this discrepancy achieves the minimum value when the two probability measures are the same. As a consequence, we may face some issues of using m-OT in practice. On the other hand, the population version of BoMb-OT defines a proper metric in the space of probability measures and it can overcome all the previous issues of m-OT. Therefore, in terms of statistical inference purposes with a minibatch optimal transport framework, BoMb-OT may offer a good alternative for m-OT. Since our focus is not to approximate the full-batch OT, Theorem 2 is indeed about the convergence property of BoMb-OT to its population version, which yields the necessary sample complexity for such approximation.
>
> From the practical viewpoint, the BoMb-OT can be used to estimate the gradient of the parameter of interest, the transportation plan, and the transportation cost with the time complexity of $\mathcal{O}(k^2m^2/\epsilon)$ and it only requires to store a cost matrix of size $m \times m$ at a time. Since we often use optimal transport as a loss to train a model (e.g. neural network) iteratively, using a cheap estimation like BoMb-OT is more convenient and economical. We would like to recall that the original full-batch OT divergence (e.g., full-batch Wasserstein, full-batch Sinkhorn,  full-batch sliced Wasserstein) cannot be used when $n$ is large, and m-OT and BoMb-OT are discrepancies that are defined by using OT with random sub-samples of size $m$. The difference between BoMb-OT and m-OT is that BoMb-OT suggests discarding pairs of mini-batches that are far away in terms of the optimal transportation metric.
>
> Finally, we would like to remark that, when $m=1$ and $d=W_2$, the population BoMb-OT reverts into the conventional OT while the population m-OT reverts to the $\mathcal{L}_2$ norm.
>
> **Q2**: How large k and m need to be so we get a reasonable approximation?.
>
> **Answer**: In practice, the choice of $m$ is limited due to the memory limitation of computational devices, e.g., $m=100$. For $k$, a bigger $k$ leads to a better estimation of the population form for BoMb-OT (See the statement of Theorem 2). We have discussed how to practically choose $k$ and $m$ in the paper in the paragraph “Choosing k and m”. In summary, $k$ and $m$ should be chosen as big as possible based on the memory of computational devices and the training time budget, and $k$ can be chosen to be larger than $m$ in the case when we have two different computational memories.

---

> ### Author Response · Authors · 2021-11-16
> **Response to Reviewer 9B4c: Part 2**
>
> **Q3**: Comparisons with true Wasserstein distance are mostly lacking.
>
> **Answer**: Thanks for your comment. We would like to emphasize that BoMb-OT is a mini-batch strategy to compare large-scale probability measures and can be thought of as a computational alternative to conventional OT distances. In particular, the original OT distances or Sinkhorn divergence need to compute and store a $n\times n$ cost matrix where $n$ is the number of data samples. When $n$ is large (e.g., millions), it is almost impossible to compute and store that matrix in our real data experiments. In contrast, BoMb-OT and m-OT only need to compute and store a $m\times m$ cost matrix at a time. Furthermore, BoMb-OT and m-OT do not change the nature of transportation plans (Sinkhorn or sliced Wasserstein), they only propose a strategy to mimic the transportation plan and the model gradient by multiple mini-batch transportation plans which are possible to solve. We would like to recall that almost all applications using OT in deep learning need to use mini-batch OT, which reinforces the impact of the new BoMb-OT. We include here some famous papers that use mini-batch OT as their loss below (in these applications, it is not possible to implement the full-batch sliced/ regularized/ Sinkhorn, etc.):
>
> 1. Deep Generative Models:
>      1. "Learning Generative Models with Sinkhorn Divergences" - https://arxiv.org/abs/1706.00292. The paper uses the gradient signal of m-OT loss (k=1) to train the deep generative network.
>      2. "Improving GANs Using Optimal Transport" https://arxiv.org/pdf/1803.05573.pdf. The paper proposes to use additional debiased terms for m-OT to yield a better objective loss.
>      3."Generative Modeling using the Sliced Wasserstein Distance" https://openaccess.thecvf.com/content_cvpr_2018/papers/Deshpande_Generative_Modeling_Using_CVPR_2018_paper.pdf. The paper uses m-OT in which the OT distance is sliced Wasserstein.
>     4. "Learning Generative Models across Incomparable Spaces" https://arxiv.org/abs/1905.05461. The paper uses m-OT with the Gromov Wasserstein as the OT distance for learning to generate images from different metric spaces.
>     5. "Sliced Wasserstein Auto-Encoders" https://openreview.net/forum?id=H1xaJn05FQ. m-OT (with sliced Wasserstein) is used to penalize the latent code distribution and the prior distribution.
> 2. Deep Domain Adaptation:
>     1. "DeepJDOT: Deep Joint Distribution Optimal Transport for Unsupervised Domain Adaptation" https://arxiv.org/abs/1803.10081. Here, m-OT is used with a domain-adaptation-specific ground metric.
>     2. "Sliced Wasserstein discrepancy for unsupervised domain adaptation" https://openaccess.thecvf.com/content_CVPR_2019/html/Lee_Sliced_Wasserstein_Discrepancy_for_Unsupervised_Domain_Adaptation_CVPR_2019_paper.html. In this paper, m-OT in which OT distance is sliced Wasserstein is utilized to be the objective loss for transfer knowledge between two domains.
> 3. Others
>     1. "Optimal Transport: Fast Probabilistic Approximation with Exact Solvers" https://arxiv.org/pdf/1802.05570.pdf. The paper proposes to use m-OT as an approximation of the original optimal transport.
>     2. "Distribution Matching for Crowd Counting" https://arxiv.org/abs/2009.13077. m-OT loss is used in the application of crowd counting.
>
> --- Finally, we would like to remark that the only application that is possible to run full-OT is the color transfer application. In this task, we consider transporting a color palette of two images of 3000 pixels hence full-OT can be run. However, when the number of pixels is one million (HD images), full-OT cannot be used. The benefit of m-OT and BoMb-OT is that they can iteratively transport a random part of the source image to a random part of the target image, e.g., 10 pixels (m=10). We have plotted the transported image of full-OT in Figure 2 in the main text of the revision of the paper. According to this figure, BoMb-OT’s image looks more similar to the full-OT’s image than m-OT.

---

> ### Author Response · Authors · 2021-11-16
> **Response to Reviewer 9B4c: Part 3**
>
> **Q4**: Another suggestion (optional): I believe this method can be useful for aligning spaces, as in https://arxiv.org/abs/1809.00013. It would be good to have comparisons there as well and to show whether this mini-batch approximation can be reasonable (i.e. compare against the no-mini batch case).
>
> **Answer**: Thank you for suggesting that reference. We have cited this paper as an example of using mini-batch OT in NLP in the revision of our paper. As discussed in Section 4.2 of your mentioned paper,  the authors wrote  “When running Algorithm 1 for the full set of embeddings is infeasible (due to memory limitations), one must decide what fraction of the embeddings to use during optimization.”. Therefore, the authors of the paper use a direct setting of mini-batch Gromov Wasserstein with $k$=1, $m$ is chosen to 20000. However, we would like to raise the natural difference between Gromov Wasserstein and Wasserstein that affects the design of a mini-batch scheme. In particular, Gromov Wasserstein distance transports between *similarity matrices* of two measures while Wasserstein distance transports their *supports*.
>
>  For greater detail, we would like to explain the challenge when using mini-batches for Gromov Wasserstein. We define two metric-measured spaces $(X^n,\mu_n,c_1)$  and $(Y^n,\nu_n,c_2)$, where $X^n =\\{x_1,\ldots,x_n\\}$, $Y^n=\\{y_1,\ldots,y_n\\}$, $\mu_n= \frac{1}{n} \sum_{i=1}^n \delta_{x_i}$ and $\nu_n= \frac{1}{n} \sum_{i=1}^n \delta_{y_i}$ are two measures on $X^n$ and $Y^n$ respectively, $c_1$ is a distance on $X^n$ and $c_2$ is a distance on $Y^n$. We define $C^X$ is the inner distance matrix (similarity matrix) of $X^n$ where $C^X_{ij} = c_1(x_i,x_j)$ and $C^Y$ is the inner distance matrix of $Y^n$ where $C^Y_{kl} = c_2(y_k,y_l)$. The Gromov Wassertein distance of order $p$ between $(X^n,\mu_n,c_1)$  and $(Y^n,\nu_n,c_2)$ is defined as:
>
>  $GW(\mu_n,\nu_n, C^X,C^Y)=  \min_{\pi \in \Pi(\mu_n,\nu_n)}  \sum_{i,j,k,l} \| C^X_{ij} - C^Y_{kl} \|^p \pi_{ik} \pi_{jl}$.
>
> A naive approach for mini-batch Gromov Wasserstein is to use the averaging scheme as m-OT.  In particular,  $X_1^m, \ldots, X_k^m$ are sampled without replacement from $\{X^n \choose m\}$, and $Y_1^m, \ldots, Y_k^m$ are sampled without replacement from $\{Y^n \choose m\}$. As in the paper, we define $P_{X^m} = \frac{1}{m} \sum_{i=1}^m \delta_{x_i}$ where $X^m = \\{x_1,\ldots,x_m\\}$. The naive mini-batch Gromov Wasserstein can be defined as:
>
> m-$GW^{k,m} (\mu_n,\nu_n, C^X,C^Y)$=$\frac{1}{k^2} \sum_{i=1}^k \sum_{j=1}^k GW(P_{X^m_i},P_{Y^m_j}, C^{X^m_i},C^{Y^m_j})$,
>
> where $C^{X^m_i}$ has entries equal $0$ except those indexed of samples that belongs to $X^m_i$.
>
> The mini-batch transportation plan of mini-batch Gromov Wasserstein can be defined as:
>
> $\hat{\pi_1}^{k,m}(\mu_n,\nu_n, C^X,C^Y)$=$\frac{1}{k^2} \sum_{i=1}^k \sum_{j=1}^k \pi^{GW(P_{X^m_i},P_{Y^m_j}, C^{X^m_i},C^{Y^m_j})}$.
>
> ***Challenges:*** For a pair of mini-batches $X_i^m$ and $Y_j^m$, $GW(P_{X^m_i},P_{Y^m_j}, C^{X^m_i},C^{Y^m_j})$ only considers the transportation sub-matrices of $C^X$ and $C^Y$, that leads to the problem of non-optimal matchings. Similar to m-OT, the above formulation treats all pairs of mini-batches the same by averaging, which cannot correct any non-optimal matchings. Moreover, while OT aligns samples by transporting directly samples, Gromov Wasserstein aligns samples by transporting *similarity matrices*. So, the extent of losing information because of using mini-batches is more serious. We would like to recall that we can apply directly the batch of mini-batches scheme here to obtain BoMb-GW:
>
> BoMb-$GW^{k,m} (\mu_n,\nu_n, C^X,C^Y)$=$\min_{\gamma \in \Pi(\mu_k^{\otimes m}, \nu_k^{\otimes m})} \sum_{i=1}^k \sum_{j=1}^k \gamma_{ij} GW(P_{X^m_i},P_{Y^m_j}, C^{X^m_i},C^{Y^m_j})$,
>
> and the transportation plan:
>
> $\hat{\pi_2}^{k,m}(\mu_n,\nu_n, C^X,C^Y)$=$\sum_{i=1}^k \sum_{j=1}^k \gamma_{ij} \pi^{GW(P_{X^m_i},P_{Y^m_j}, C^{X^m_i},C^{Y^m_j})}$,
>
> where $\mu_k^{\otimes m} = \frac{1}{k} \sum_{i=1}^k \delta_{X_i^m}$ and $\nu_k^{\otimes m} = \frac{1}{k} \sum_{j=1}^k \delta_{Y_j^m}$.
>
>
> However, because of the natural difference of Gromov Wasserstein from OT, the effect of the additional OT in BoMb-GW is not trivial as in BoMb-OT where the transportation type in both levels is OT.
>
> Therefore, it still remains a challenge to create a good mini-batch scheme for Gromov Wasserstein. Hence, we leave that direction for future work.

---

> ### Author Response · Authors · 2021-11-28
> **Response to Reviewer 9B4c - Any further questions on our current draft**
>
> We would like to thank you again for your reviews and feedback. We have updated our manuscript and added replies to your comments and questions.
>
> Given that your current score is 5, we would appreciate it if you could let us know if our responses have addressed your concerns and whether you still have any other questions on the current draft.
>
> We would be happy to do any follow-up discussion or address any additional comments.

---

### Author Response · Authors · 2021-11-22
**Look forward to your final feedback**

Dear reviewers,

Thank you for spending your time evaluating our paper and providing comments.

We look forward to hearing your feedback regarding our rebuttals and the revision of our paper. We are happy to discuss if you still have any other concerns.

Best,

Authors

---

### Decision · Program_Chairs · 2022-01-20

**Decision:**

Reject

**Comment:**

This paper propose a way to make minibatch Optimal transport (m-OT) more efficient by computing an optimal assignment (in the OT sens) and us this assignment to compute instead a hierarchical OT loss (bomb-OT ) that can be used instead of the m-OT loss. The authors discuss how the equivalent OT plan with bomb-OT is much more sparse, and how the proposed approach is actually not biased when the number of mini-batches $k\rightarrow \infty$ . Numerical experiments  show that the proposed method allows a gain in performances in applications such as generative modeling, domain adaptation, color transfer and approximate Bayesian computation.

The paper originally got borderline-negative scores from the reviewers. While the reviewers acknowledged that the idea is interesting, they had some concerns about the theoretical results strength, some missing baselines and discussions in the numerical experiments. The authors did a detailed reply that clarified some problems. the new numerical experiments with m-UOT were also greatly appreciated by the reviewers but they also raised some questions about the paper. Some concerns detailed below about the comparison with m-OT appeared during the reviewers discussion. Despite the new information,  the reviewers reached an agreement that this paper is interesting but needs more work and another round of reviews before acceptance. For theses reasons  the AC recommends a rejection for this paper.

More details and suggestions below:

- While it is clearly not the objective of the paper a discussion about the proximity of the average plan to the exact OT plan is interested. Also a short numerical experiments showing that the bomb-OT average plan is closer to the exact plan than m-OT would be a good illustration of the better performance of bomb-OT. This seems more important for the paper than the color transfer experiments that is kind of a toy problem.

- After checking the definition in the paper and discussion between reviewers it appeared that the comparison with m-OT is a bit unfair due to the reformulation of the problem in (1). indeed in the usual formulation, k pairs of independent  minibatches are used and the OT is done on those pairs (a sum of k OT) not on all the possible pairwise permutation as in definition of m-OT in equation (1). In other  words in m-OT the batches are supposed to be independent which is not the case  in the proposed formulation (it is equivalent in the population case though).  It means that in practical application, for the same computational complexity  (k^2 OT computed), m-OT actually uses $k^2m$ independent samples on each distribution  whereas the bomb-OD (and the m-OT defined in equation (1) ) use $km$ samples . By implementing m-OT as  in (1) they actually prevent m-OT to explore the dataset as its original  formulation does. This means that all the experiments should be done either with the original m-OT implementation of both the original and (1) in addition to bomb-OT. The proposed method will proably work better but the current experiment do not allow this fair comparison.

- The theoretical result need more discussion and justification.  For instance  m-OT converges to its population value in
 $O(m^{1/2}n^{-1/2}+k^{-1/2})$ that is independent from the dimensionality  $d$, but the authors prove the  concentration of bomb-OT in  of $O(m^{1/2}n^{-1/d})$  which is  clearly a problem for large $d$. Also the dependence on  $k$ of the convergence would be important since  bomb-OT is well defined is true only in the population case where $k$ is  large. Note that the claim that it is well defined and hence better is also a  bit dubious because it is well defined for $k=\infty$, which is also the case for m-OT when $m=\infty$. Both $m$ and $k$ large will lead to not practical optimization problems so they are comparable except that m-OT converged to the true OT plan when $m\rightarrow \infty$ which is not the case for bomb-OT.

- While the contribution of the paper in indeed a methodological method and does not require to be state of the art on all applications the numerical experiments should be improved. First as discussed above the comparison with m-OT is actually unfair an do not correspond to what in done in practice (where all mini batches are independent). m-OT should be implemented with  $k^2$ truly independent minibatches.

- Second , the authors use approximate W2 on two of  the GAN dataset and FID on the third. This is  problem because approximate W2 is not defined in the paper. FID is the standard performance measure and should be used for all dataset.

- Third the novel experiments comparing also raises a lot of questions. m-UOT is far better than BoMb-OT suggesting that Unbalanced OT can compensate for the limits of m-OT far better than bomb-OT itself. Yes there is a slight increase in performance  for ebomb-UOT over m-UOT but is is so small (0.08 %) that it is hard to find them significant, especially since we have no variance. This result that is provided only for DA application actually  suggest that the competitor of bomb-OT is m-UOT and not m-OT so it should also be part of the comparison in the other experiments. The authors talk in their replay about the limits of m-UOT but stating that the experiences are not done in the paper is not an excuse for evaluating this clear competitor on other problems and showing numerically these limits.

- Finally in the current version of the paper puts a lot of things in the annex that make the paper clearly not self content. Some experiments could go in annex/supp for instance the color transfer to make place for more details in the main paper.

Note that it is not one of those comments above that lead the the reject decision but the sum of them that clearly show that the paper needs more work.